# Stem Leydig cells support macrophage immunological homeostasis through mitochondrial transfer in mice

Ani Chi [1,2,3,10], Bicheng Yang[1,10], Hao Dai[2], Xinyu Li[1], Jiahui Mo[1], Yong Gao[4], Zhihong Chen[1], Xin Feng[1], Menghui Ma[5], Yanqing Li[5], Chao Yang[2], Jie Liu[2], Hanchao Liu[1], Zhenqing Wang[1], Feng Gao[1,9], Yan Liao[6], Xuetao Shi[2,7,8] ✉, Chunhua Deng [1] ✉ & Min Zhang[1,3] ✉

As testicular mesenchymal stromal cells, stem Leydig cells (SLCs) show great promise in the treatment of male hypogonadism. The therapeutic functions of mesenchymal stromal cells are largely determined by their reciprocal regulation by immune responses. However, the immunoregulatory properties of SLCs remain unclear. Here, we observe that SLCs transplantation restore male fertility and testosterone production in an ischemia–reperfusion injury mouse model. SLCs prevent inflammatory cascades through mitochondrial transfer to macrophages. Reactive oxygen species (ROS) released from activated macrophages inducing mitochondrial transfer from SLCs to macrophages in a transient receptor potential cation channel subfamily member 7 (TRPM7)-mediated manner. Notably, knockdown of TRPM7 in transplanted SLCs compromised therapeutic outcomes in both testicular ischemia–reperfusion and testicular aging mouse models. These findings reveal a new mechanism of SLCs transplantation that may contribute to preserve testis function in male patients with hypogonadism related to immune disorders.

Testicular function relies on a homeostatic interstitial microenvironment that is essential for continuous spermatogenesis[1]. Tissue-resident mesenchymal stromal cells serve as sensors of damage and actively communicate with the tissue microenvironment, especially inflammatory components[2]. Stem Leydig cells (SLCs) are tissue-specific multipotent stromal cells that reside in the testicular interstitium[3]. We have demonstrated that SLCs support long-term homeostasis by regenerating new Leydig cells (LCs) after transplantation into an LC-disrupted or senescent mouse model[4,5]. However, only a few autogenous SLCs can differentiate into LCs, and the increase in serum testosterone levels only 4-8 weeks after transplantation in aging nonhuman primate models is characterized by a low-grade inflammatory response[6]. It is well accepted that the reparative functions of mesenchymal stromal cells rely on their interaction with the

[1]Department of Andrology, The First Affiliated Hospital, Sun Yat-sen University, Guangzhou 510080, China. [2]School of Materials Science and Engineering, South China University of Technology, Guangzhou 510640, China. [3]Institute of Precision Medicine, The First Affiliated Hospital of Sun Yat-sen University, Guangzhou 510080 Guangdong, China. [4]Reproductive Medicine Center, The Key Laboratory for Reproductive Medicine of Guangdong Province, The First Affiliated Hospital of Sun Yat-sen University, Guangzhou, Guangdong 510080, China. [5]Center of Reproductive Medicine, the Sixth Affiliated Hospital, Sun Yat-sen University, Guangzhou 510655, China. [6]Key Laboratory of Biomedical Engineering of Guangdong Province, South China University of Technology, Guangzhou 510006, P. R. China. [7]National Engineering Research Centre for Tissue Restoration and Reconstruction and Key Laboratory of Biomedical Engineering of Guangdong Province South China University of Technology, Guangzhou 510640, China. [8]Shenzhen Beike Biotechnology Co., Ltd., Shenzhen 518054, China. [9]Present address: Reproductive Medicine Center, The Key Laboratory for Reproductive Medicine of Guangdong Province, The First Affiliated Hospital of Sun Yat-sen University, Guangzhou, Guangdong 510080, China. [10]These authors contributed equally: Ani Chi, Bicheng Yang. ✉e-mail: shxt@scut.edu.cn; dengchh@mail.sysu.edu.cn; zhangm287@mail.sysu.edu.cn

immune microenvironment in many tissues. For example, cardiac Nestin[+] mesenchymal stromal cells enhance the healing of ischemic hearts by promoting macrophage polarization[7]. However, whether SLCs possess broad immunoregulatory properties for the local microenvironment has not been investigated. Testicular resident TCF21[+] mesenchymal stromal cells are analogous to SLCs that can be activated to support LC regeneration upon injury or aging. Notably, intercellular crosstalk occurs between TCF21[+] mesenchymal stromal cells and macrophages[8]. Although the mechanism has not yet been fully elucidated, this evidence indicates that SLCs may be potential testis-specific microenvironmental immunomodulators. Therefore, it is imperative to clarify the mechanism of the reciprocal regulation of SLCs and testicular immunity to explore novel therapeutic targets for the enhancement of SLC-based cell therapy.

As testicular primary innate immunocytes, macrophages play crucial roles in testis development, tissue repair and the pathophysiological process of testicular aging[9,10]. Cell–cell communications between macrophages and interstitial cells contribute to a favorable state for germ cell development. Under homeostatic conditions, resident macrophages are closely associated with LCs, regulate steroidogenesis and contribute to the spermatogonial niche[11]. In addition, macrophages are key regulators of innate immunity responsible for acute inflammation and chronic inflammation[12]. When tissue injury occurs, macrophages are activated, which initiates a range of actions, including chemotaxis, phagocytosis, ROS production and proinflammatory factor production. Aging-associated alterations in testicular somatic cells are also characterized by the upregulation of inflammation-induced genes[13]. Mitochondria lie at the heart of the immune response[14]. There is evidence that acute and chronic inflammatory responses are associated with macrophage mitochondrial dysfunction[15,16]. However, the mechanisms by which testicular macrophages regulate tissue homeostasis in such an inflammatory niche need further exploration.

Here, by using an acute injury mouse model of testicular torsion, we discovered that SLCs can quickly sense damage signals and establish connections with resident macrophages via nanotubes, through which mitochondria are transferred from SLCs to neighboring activated macrophages. This process is regulated by TRPM7, a pattern recognition receptor expressed on SLCs. Mitochondrial propagation attenuated the inflammatory profile of macrophages. By reshaping the interstitial microenvironment, SLCs promoted LCs regeneration and restored spermatogenesis and fertility. Mechanistically, suppression of TRPM7 with short hairpin RNA (shRNA) reduced intercellular mitochondrial transfer and worsened therapeutic efficacy. Notably, these mechanisms were reproduced in chronic inflammation during testicular aging. In summary, we are the first to reveal that mitochondria transfer is a robust mechanism by which SLCs maintain immune microenvironment homeostasis in the testis, which is a novel therapeutic target for limiting inflammation in immune-related diseases associated with male infertility.

## Results

### SLCs prevent fertility injury in an acute testicular ischemia–reperfusion model

To investigate the potential immunoregulatory functions of SLCs, we established an acute injury testicular mouse model, the testicular torsion mouse model, to elucidate the properties of SLCs in response to acute injury. As a starting point, SLCs were isolated from 7-day-old mouse testes by flow cytometry sorting using CD51[+], a marker of SLCs (Supplementary Fig. 1a). After 7 days of in vitro culture, the isolated primary SLCs proliferated and formed small floating clonal spheres (Supplementary Fig. 1b). SLCs are tissue-specific mesenchymal stromal cells that reside in the testicular interstitium, and bone marrow-derived mesenchymal stromal cells (BMSCs) are widely used as therapeutics for organ failure[17] and acute and chronic disorders[18]. Local injection of mesenchymal stem cells has been shown to alleviate germ

cell injury in a testicular torsion model[19]. Therefore, we compared the reparative and homeostatic maintenance functions of SLCs and BMSCs in this testicular acute injury model. BMSCs were isolated[20] and purified by CD51. The surface markers on BMSCs were Sca-1, CD90 and CD106, and the BMSCs were negative for c-kit, CD45, and CD11b (Supplementary Fig. 1c, d).

Equal amounts of BMSCs and SLCs were transplanted into the testicular interstitium. After 28 days, testes were collected for further analysis (Fig. 1A and Supplementary Fig. 1e). We found that the testes were atrophied in the saline group and BMSCs group compared to the SLCs or sham group (Fig. 1B). Histopathological analyses showed that the structures of the seminiferous tubules and interstitium were abnormal, while SLCs transplantation significantly improved testicular structural integrity (Fig. 1B). The testis weight ratio of the whole body weight and germ cell layer were significantly improved in the SLCs group compared with the BMSCs and saline groups (Fig. 1C, D). Furthermore, staining with the specific marker of germ cell VASA and the specific marker of spermatogenesis synaptonemal complex protein 3 (SCP3) revealed high degrees of recovery of germ cells in the SLCs group (Fig. 1E–G). Sperm production analysis was analyzed by staining the tissues with PNA, a marker of sperm or haploid spermatids, which specifically binds to the acrosomes of sperm. α-SMA, which is a marker of peritubular myoid cells, can be used to indicate seminiferous tubule boundaries. Imaging revealed the disappearance of PNA[+] in the saline group, whereas the number of PNA[+] cells was significantly increased in the SLCs group compared to the BMSCs groups and saline groups (Fig. 1H, I). Epididymal sperm characteristics, including sperm counts and sperm motility, were significantly higher in the SLCs group than in the BMSCs and saline groups (Fig. 1J, K and Supplementary Fig. 1f). The epididymides from the SLCs group contained massive numbers of long sperm (Fig. 1L). To further investigate whether SLCs treatment could impact the quality of sperm, in vitro fertilization (IVF) was performed using the sperm of mice in different groups. We found a significantly increased rate of 2-cell embryos and blastocysts (88.18% and 92.78% in the sham group, 46.95% and 37.30% in the BMSCs group, and 75.00% and 53.08% in the SLCs group), while embryos failed to develop into blastocysts in the saline group (Fig. 1M and Supplementary Fig. 1g, h).

To further investigate mouse fertility, a male mouse was mated with a 3-month-old female mouse at a ratio of 1:2 (Fig. 1N). No litters were born in the saline group within 4 months, while the numbers of pups and litters were higher in the SLC group than in the BMSCs group (Fig. 1O, P). These results indicated that SLCs have superior therapeutic potential than BMSCs, which restore histological integrity and improve spermatogenesis and male fertility in testicular torsion models.

### Transplanted SLCs restore testosterone production mainly by accelerating endogenous LC recovery

Spermatogenesis is largely dependent on testosterone levels[21]. To validate whether the transplanted BMSCs and SLCs contributed to restoring hormone secretion function, testosterone levels were measured at 28 days after BMSCs or SLCs treatment. Testosterone levels were consistently higher in the SLCs group than in the BMSCs group and saline group (Fig. 2A). LCs contribute to the highest proportion of testosterone produced in men[22]. Therefore, we quantified the proportion of LHR-positive LCs in the testes of the BMSCs and SLCs groups. The ratio of LCs was analyzed by flow cytometry at 1, 3, 7 and 28 days after cell transplantation. The results showed that the proportions of LCs decreased constantly among the whole testis at 1, 3 and 7 days after testicular torsion. SLCs treatment significantly attenuated the declining trend compared to the saline and BMSCs groups (Supplementary Fig. 2a–c). Importantly, on day 28, the recovery of LCs was more significant in the SLCs group than in the saline group and BMSCs group (Fig. 2A, B). The dynamics of the LCs ratio was shown (Supplementary Fig. 2d).

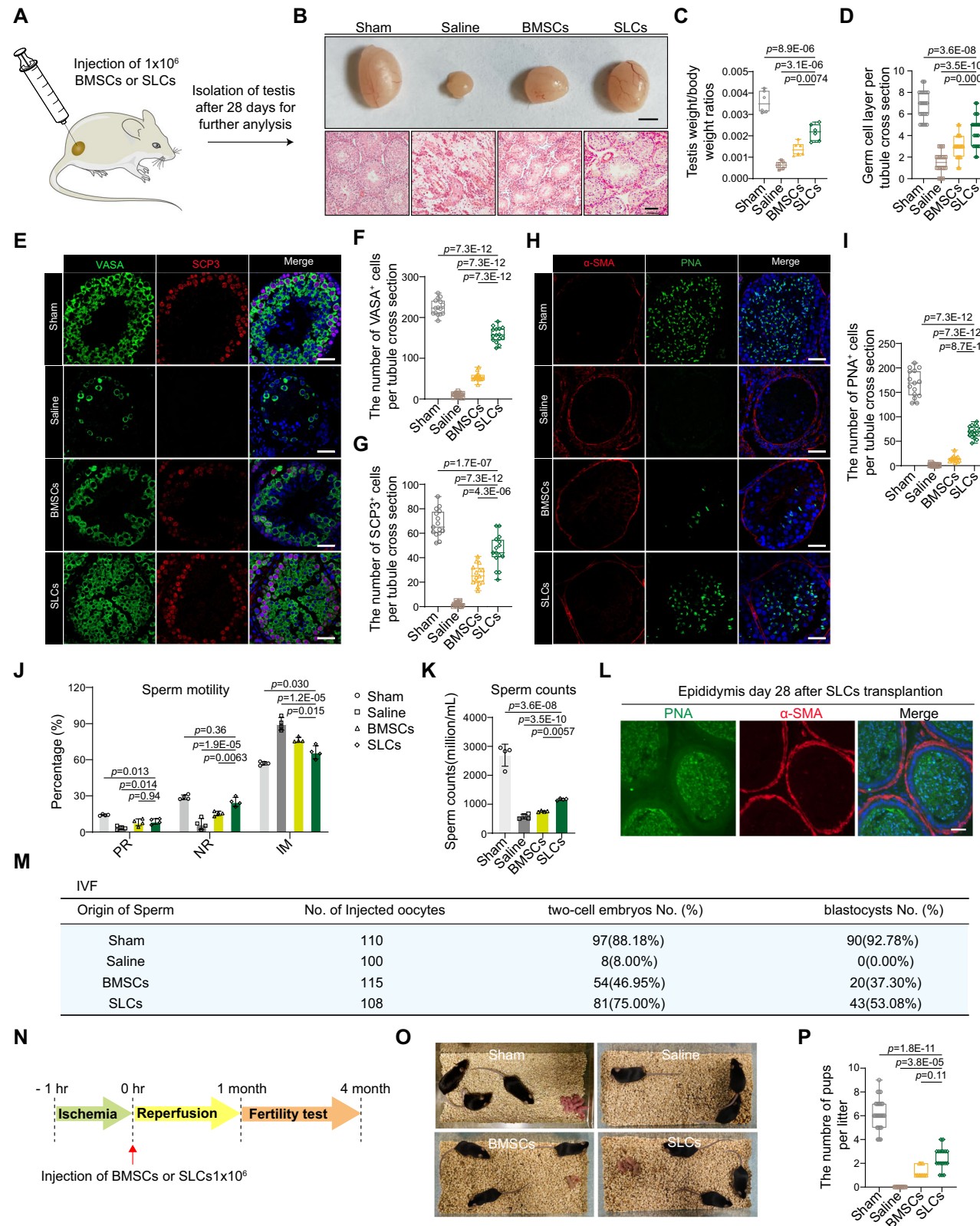

To independently validate how the exogenous transplanted cells contributed to LCs regeneration, we tracked the fate of BMSCs and SLCs inside the testes after transplantation. BMSCs$^{zsGreen}$ and SLCs$^{zsGreen}$ were sorted from actin-zsGreen mice and injected into the testicular interstitium after testicular torsion (Fig. 2D). In vivo fluorescence imaging did not reveal the migration of cells to other organs, such as the lungs, kidneys, or livers (Fig. 2E). The transplanted cells were exclusively localized to the testicular interstitium (Fig. 2F). There was a gradual decrease in the proportions of BMSCs$^{zsGreen}$ and SLCs$^{zsGreen}$ from day 1, while the percentage of SLCs$^{zsGreen}$ colonization was significantly higher than that of BMSCs$^{zsGreen}$. Unexpectedly, there was a marked decrease in abundance in the testes on day 7, and only 0.32% of SLCs$^{zsGreen}$ remained (Fig. 2G, H and Supplementary Fig. 3a). These results suggested that most transplanted cells had disappeared. If the

**Fig. 1 | Therapeutic effects of SLCs versus BMSCs on testicular function after ischemia–reperfusion. A** Schematic illustration of cell injection into the testicular interstitium. **B** Bright field diagram of testicular size (scale bar, 2 mm) and H&E staining of testis samples (scale bar, 50 μm) from the sham operation group (sham), normal saline injection group (saline), bone marrow-derived mesenchymal stromal cell (BMSCs) injection group and stem Leydig cell (SLCs) injection group. **C, D** Quantification of the testis weight/whole body weight and number of germ cell layers, n = 6 biological replicates for each group. Data are presented as the means ± SD, one-way ANOVA was used. **E** Immunostaining of VASA (green) and SCP3 (red) sections. Scale bars, 50 μm. **F, G** Quantitative analysis of VASA⁺ or SCP3⁺ cells. Data are presented as the means ± SDs. n = 6 biological replicates, one-way ANOVA. **H** Immunostaining of α-SMA (red) and PNA (green) in paraffin sections. Scale bar, 50 μm. **I** Quantitative analysis of PNA⁺ cells. Data are presented as the means ± SDs. n = 3 biological replicates, one-way ANOVA. **J, K** Quantitative analysis of sperm motility and sperm counts. PR progressive motile, NP nonprogressive motile, IM immotility. Data are presented as the means ± SDs, n = 4 biological replicates for each group. One-way ANOVA. **L** Immunostaining of α-SMA (red) and PNA (green) in the epididymis. Scale bar, 50 μm. **M** The trilinear table shows all the embryo injection, two-cell embryo and blastocyst data. **N** Timeline of the mating assessment. **O** Offspring of the different groups. **P** Quantitative analysis of the number of pups per litter. Data are presented as the means ± SDs, n = 6 biological replicates for each group, one-way ANOVA. All box-and-whisker plots denote the maximum (top whisker), 75th (top edge of box), 25th (bottom edge of box), and minimum (bottom whisker) percentiles and the median (line in box). Source data are provided as a Source Data file.

surviving transplanted SLCs^zsGreen differentiated into LCs, zsGreen-positive LCs would be detected in the testes. On day 28, we found that the LHR⁺ cells in the BMSCs^zsGreen group were zsGreen negative, while approximately 90% of LHR⁺ cells were zsGreen negative and only 10% of LHR⁺ cells were zsGreen positive in the SLCs^zsGreen group (Fig. 2I and Supplementary Fig. 3b). The ratio of LHR⁺zsGreen⁻ cells was significantly higher than that of LHR⁺zsGreen⁺ cells (Fig. 2J), indicating that a small number of LCs were derived from SLCs^zsGreen. We isolated LHR⁺zsGreen⁺ cells and LHR⁺zsGreen⁻ cells by flow cytometry. Subsequently, we collected the culture supernatant for testosterone determination analysis. Notably, we found that there were no differences in testosterone secretion by LHR⁺zsGreen⁺ or LHR⁺zsGreen⁻ cells (Fig. 2K). Immunofluorescence staining revealed that HSD3β⁺zsGreen⁺ or CYP11A1⁺zsGreen⁺ cells were found in the testicular interstitium in the SLCs^zsGreen group (Fig. 2L, M). These results suggested that only a few exogenous SLCs differentiated into LCs, which is not a major contributor to the recovery of testosterone production. In contrast, endogenous SLCs played a more critical role in the recovery of testosterone production and the regeneration of LCs. We further investigated whether cell numbers affect the regeneration of LCs, and different doses of SLCs^zsGreen ranging from 5 × 10⁵ to 2 × 10⁶ were transplanted into the testicular interstitium (Supplementary Fig. 4a). After 28 days, we found that testes were atrophied in the 5 × 10⁵ and 2 × 10⁶ groups compared to the 1 × 10⁶ group (Supplementary Fig. 4b), and the number of LHR⁺ (the total LC population) and LHR⁺zsGreen⁺ (the exogenous population) cells in the whole testis were also the highest in the 1 × 10⁶ SLCs groups (Supplementary Fig. 4c, d). The 1 × 10⁶ SLCs transplantation exerted the best effect on the regeneration of LCs. These results indicated a dose dependence of the therapeutic potential of SLCs, which showed unsatisfactory results with lower and higher doses.

To further confirm LCs regeneration mainly from endogenous SLCs, we investigated the self-renewal and differentiation capacities of endogenous SLCs (Supplementary Fig. 5a). Endogenous SLCs were sorted on day 28 in the Actin-zsGreen mouse testicular torsion model, and single-cell sphere formation assays revealed that the seeding of single cells exhibited higher clonogenic efficiency in the SLCs group than in the BMSCs group and saline group (Supplementary Fig. 5b, c). Accordingly, endogenous SLCs in the SLCs group showed a higher efficiency of differentiation into mature LCs, producing more testosterone (Supplementary Fig. 5d) and expressing higher levels of LCs markers, such as *Cyp11a1*, *Lhr*, *HSD3β* and *Star* (Supplementary Fig. 5e). These findings implied that implanted SLCs improve the regeneration of LCs by promoting endogenous SLCs differentiation.

### Transplanted SLCs suppress the inflammatory cascade response in the early stage

Testicular torsion-induced damage is a typical ischemia–reperfusion injury that involves the initiation of inflammatory responses. SLCs differentiation is largely dependent on the niche cell. Previous studies have reported that inflammation leads to lower testosterone production by affecting SLCs proliferation and differentiation²³. Therefore, we investigated whether the transplanted cells promoted endogenous LCs recovery by effectively dampening the excessive immune response in the early stage (days 1, 3 and 7) (Fig. 3A). The obtained results indicated no significant difference in the CD45⁺F4/80⁺ macrophage ratio on day 1 (Supplementary Fig. 6a–c), while the proportions of macrophages were most markedly decreased among inflammatory cells on day 3 in the SLCs group compared with the BMSCs and saline groups (Fig. 3B, C). Similarly, immunofluorescence staining analysis demonstrated that the number of CD45⁺F4/80⁺ macrophages in the testes was largely reduced in the SLCs group on day 3 (Supplementary Fig. 6d). On day 7, the macrophages showed a decreasing trend, indicating a period of inflammation healing (Supplementary Fig. 6e, f). The gene expression levels of the proinflammatory cytokines IL-1α, IL-1β and tumor necrosis factor-α (TNF-α) in macrophages were suppressed in the SLCs group compared to the BMSCs group (Fig. 3D) on day 3. Consistent with this finding, cell apoptosis of the total testis was downregulated in the SLCs group compared to the BMSCs and saline groups (Fig. 3E–G and Supplementary Fig. 7a). Ischemia–reperfusion drives the influx of inflammatory leukocytes, including classically monocytes and neutrophils²⁴. On days 1, 3 and 7, flow cytometry analyses of peripheral blood cells indicated significantly reduced numbers of CD11b⁺Ly6C^high monocytes (Fig. 3H, I and Supplementary Fig. 7b) and CD11b⁺Ly6G^high neutrophils (Fig. 3J, K and Supplementary Fig. 7c) in the SLCs group compared to the BMSCs group. These outcomes were more pronounced on day 3. Furthermore, SLCs transplantation significantly reduced the CD11b⁺Ly6C^high monocyte (Supplementary Fig. 8a–c) and CD11b⁺Ly6G^high (Supplementary Fig. 8a, d, e) neutrophil numbers in injured testes compared to the BMSCs group and saline group on days 1, 3 and 7. In summary, based on these findings, we concluded that transplanted SLCs suppress the inflammatory cascade response in the early stage.

Macrophages are critical for tissue repair, and mesenchymal stromal cells affect macrophage polarization. We next investigated the effect of the transplanted SLCs on the polarization of macrophages. Among all tissue-infiltrating macrophage populations, we observed suppressed infiltration of CD45⁺F4/80⁺CD86⁺ or CD45⁺F4/80⁺iNOS⁺ M1 proinflammatory macrophages (Supplementary Fig. 9a, b) and enhanced recruitment of CD45⁺F4/80⁺CD206⁺ or CD45⁺F4/80⁺Arg1⁺ anti-inflammatory M2 macrophages in the SLCs group compared with the BMSCs group (Supplementary Fig. 9c, d). These results suggested that SLCs regulated the inflammatory properties of macrophages and created a favorable regenerative immune microenvironment for tissue repair.

### SLCs transfer mitochondria to resident macrophages through nanotubes after ischemia–reperfusion

To further explore the abovementioned immunomodulatory mechanism, we compared the biological properties of BMSCs and SLCs by RNA sequencing (RNA-seq). The volcano map shows marked

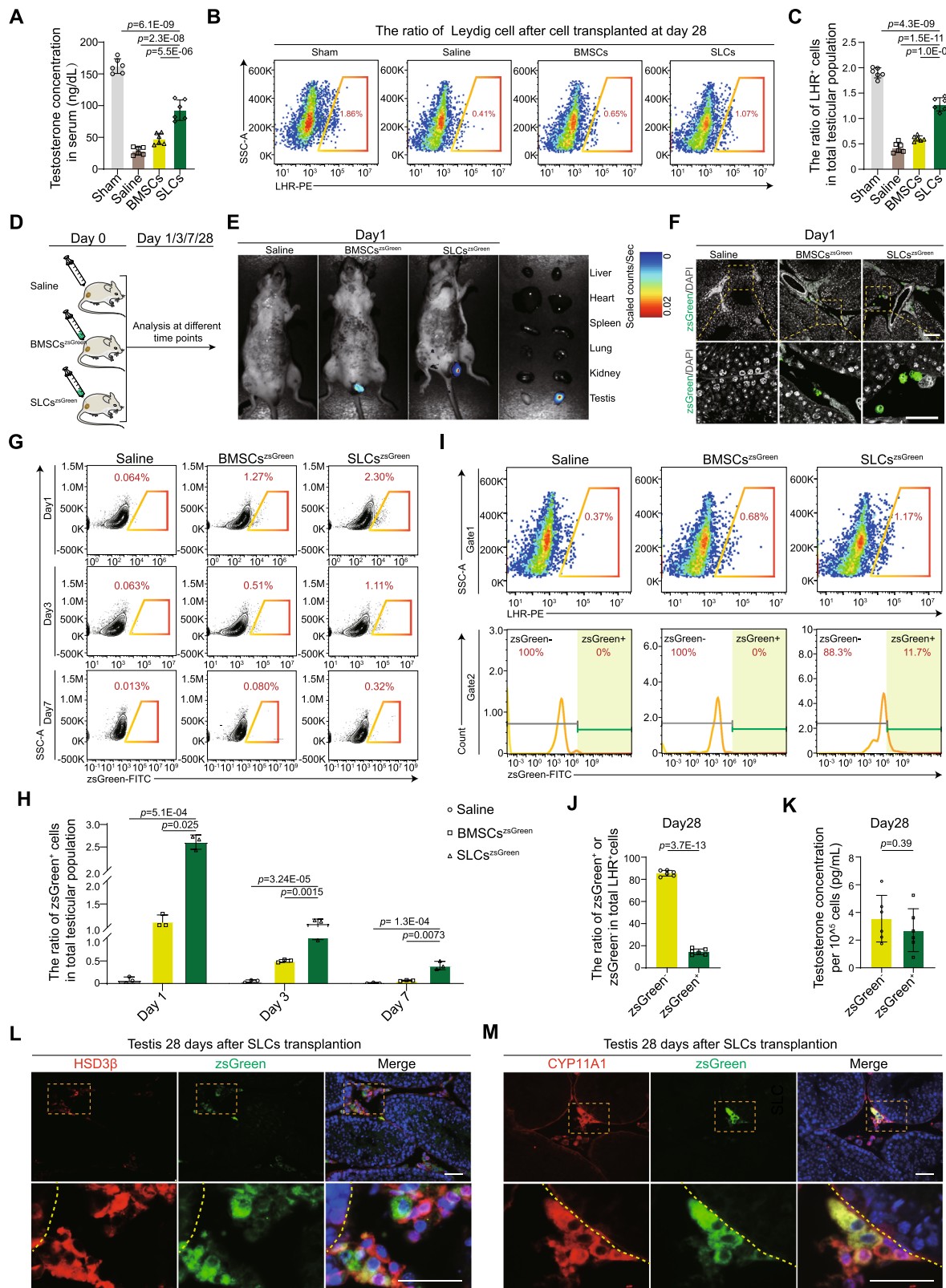

variations in the transcriptome profiles of BMSCs and SLCs. There were 3239 downregulated and 3544 upregulated differentially expressed genes (DEGs) between BMSCs and SLCs (Fig. 4A). Gene Ontology (GO) analysis of the upregulated genes showed that they were enriched in processes such as "regulation of microtubule polymerization or depolymerization" and "regulation of protein-containing complex assembly". Notably, genes upregulated in SLCs were enriched in

"leukocyte homeostasis" processes, suggesting a possible mechanism lying between SLCs and immune responses (Fig. 4B). In detail, the expression of genes related to macrophage recognition[25], activation, migration, and the inflammatory response was significantly elevated (Fig. 4C). To determine the exact effect of SLCs on macrophage function, we set up a simple experiment coculturing SLCs and macrophages. Interestingly, we observed that SLCs and macrophages

**Fig. 2 | A very small number of transplanted SLCs differentiate into LCs.**
**A** Quantitative analysis of the serum testosterone levels in the different groups. Data are presented as the means ± SDs, n = 6 biological replicates for each group, one-way ANOVA. **B** Flow cytometry detection of the percentage of LHR[+] cells among total testicular cells. **C** Quantitative analysis of the percentage of LHR[+] cells among the total testicular cells. Data are presented as the means ± SDs, n = 6 biological replicates for each group, one-way ANOVA. **D** Schematic of the experimental procedure used for BMSCs[ZsGreen] or SLCs[ZsGreen] transplantation. **E** BMSCs[ZsGreen] or SLCs[ZsGreen] were traced with an in vivo fluorescence imaging system. **F** Representative images of BMSCs[ZsGreen] and SLCs[ZsGreen] in the interstitial area. Scale bar, 50 μm. **G** Flow cytometry for detecting the percentage of BMSCs[ZsGreen] or SLCs[ZsGreen] among the total testicular cells on day 1, 3 and 7. **H** Quantitative analysis

of the percentage of BMSCs[ZsGreen] or SLCs[ZsGreen] on days 1, 3 and 7. Data are presented as the means ± SDs, n = 3 biological replicates for each group, one-way ANOVA. **I** Flow cytometry for detecting the percentage of LHR[+]zsGreen[+] or LHR[+]zsGreen[-] cells among total testicular cells on day 28. **J** Quantitative analysis of the percentage of LHR[+]zsGreen[+] or LHR[+]zsGreen[-] cells on day 28 after testicular torsion. Data are presented as the means ± SDs. n = 6 biological replicates for each group, unpaired two-tailed Student's t test. **K** Quantitative analysis of testosterone production in ZsGreen[+]LHR[+] and ZsGreen[-]LHR[+] cells in vitro. Data are presented as the means ± SDs. n = 6 biological replicates for each group, unpaired two-tailed Student's t test. **L, M** Immunofluorescence staining showing ZsGreen[+] cells (green) costained with HSD3β (red) or CYP11A1 (red). Scale bar, 50 μm. Source data are provided as a Source Data file.

physically connected via nanotube-like structures at 24 h (Supplementary Fig. 10a, b), and scanning electron microscopy (SEM) further confirmed these results (Supplementary Fig. 10c). Coincidentally, "regulation of microtubule polymerization or depolymerization" signal transduction was upregulated in SLCs, which is a key regulator of nanotube formation[26]. Nanotube structures have been reported to mediate intercellular trafficking of proteins and organelles, such as alpha-synuclein[27] and mitochondria[28]. Mitochondrial function is vital for the regulation of the activation, differentiation, and survival of macrophages[29]. Mitochondrial transfer has been shown to help donor mitochondria integrate into the endogenous mitochondrial network of receptor cells and change their bioenergetic status[30]. Therefore, we determined whether nanotubes mediated the transfer of mitochondria between SLCs and macrophages. Primary macrophages sorted from testes 1 day after testicular torsion were labeled with CellTrace Violet. BMSCs and SLCs were labeled with the red fluorescent probe Mito-Tracker Red CMXRos and cocultured with macrophages for 24 h (Fig. 4D). The mitochondrial transfer rate was quantified by flow cytometry, and the gate selection strategy is presented (Supplementary Fig. 10d, e). MitoTracker-positive macrophages indicated that macrophages had received mitochondria from BMSCs or SLCs. Remarkably, we observed that SLCs had a higher mitochondrial transfer rate than BMSCs (Fig. 4E, F). In addition, the mitochondrial mass of SLCs was higher than that of BMSCs (Supplementary Fig. 11a–c). Electron macrographs showed that SLCs had more mitochondria (Supplementary Fig. 12d, e), which were longer in length but not significantly different in width when compared to mitochondria in BMSCs (Supplementary Fig. 11f, g).

Studies have reported that nanotubes, macrovesicles and gap junctions are the main routes that mediate intercellular mitochondrial transport[31]. Thus, we further investigated whether there were other possible mechanisms involved in the mitochondrial transfer from SLCs to macrophages. Macrophages were cocultured with SLCs in a transwell system that allowed the exosomes to pass through but prevented direct cellular communication via physical contact (Supplementary Fig. 12a). Intriguingly, mitochondrial transfer was noted only in direct contact conditions, excluding the exocytosis pathway (Supplementary Fig. 12b, c). Then, the coculture system was supplemented with Gap27 (a blocker of gap junctions), cytochalasin D (a blocker of actin polymerization), or nocodazole (a blocker of microtubule polymerization). Cytochalasin D or nocodazole significantly decreased the mitochondrial transfer rate, while Gap27 treatment slightly decreased it (Supplementary Fig. 12d–g). In detail, confocal imaging showed that mitochondria from the SLCs were transferred to macrophages and finally internalized into macrophages (Fig. 4G). In addition, some neighboring macrophages were tightly attached to the SLCs (Supplementary Fig. 12h). To rule out the leakiness of the dye, SLCs were transfected with the Mito-DsRed lentivirus and then cocultured with macrophages. Confocal imaging showed that mitochondria from the SLCs were transferred to macrophages through the F-actin-positive and α-tubulin-positive nanotubes (Fig. 4H).

We then assessed whether SLCs transferred mitochondria to macrophages as we observed in vivo. BMSCs and SLCs were transfected with the Mito-DsRed lentivirus and then transplanted into testes after testicular torsion. Immunofluorescence staining showed the presence of Mito-DsRed mitochondria in F480[+] macrophages (Fig. 4H). Flow cytometry analysis revealed that after 24 h, Mito-DsRed levels in CD45[+]F480[+] macrophages were higher in the SLCs group than in the BMSCs group (Fig. 4I, J and Supplementary Fig. 13a, b). Taken together, these results showed that SLCs establish a cellular network with macrophages, through which mitochondria are transferred to neighboring activated macrophages.

**SLCs sense ROS and stimulate mitochondrial transfer in a TRPM7-dependent manner**
Mesenchymal stromal cells sense mitochondrial damage-associated molecular patterns (DAMPs) released from injured cells as danger signals[32]. Physiologically, DAMPs are recognized by pattern recognition receptors (PRRs), including cytosolic DNA sensors (CDSs), G-protein-coupled receptors (GPCRs), ion channels, NOD-like receptors (NLRs), Toll-like receptors (TLRs), and triggering receptors expressed on myeloid cells (TREMs)[33]. However, it has not been established which receptor on SLCs regulates mitochondrial transfer. To answer this question, we assessed the expression of all PRRs on SLCs by RNA-seq. Among all the receptors, TRPM7 is one of the most highly expressed in SLCs compared to BMSCs, which highlights TRPM7 as the top candidate involved in sensing damage signals (Fig. 5A). TRPM7 protein expression was confirmed by western blotting (Supplementary Fig. 14a, b). Immunofluorescence staining revealed that SLCs were localized in the interstitium and expressed Nestin and TRPM7 (Supplementary Fig. 14c). As reported previously, intercellular mitochondrial transport through nanotubes requires a motor/adaptor/receptor protein complex that includes motor proteins (KIF5), adaptor proteins (RHOT1 and RHOT2), and trafficking kinesin proteins (TRAK1 and TRAK2)[34]. By protein–protein interaction analysis with the STRING Consortium database, TRPM7 was identified as a key gene within the protein complex (Supplementary Fig. 14d). Consistent with these results, the relationship between TRPM7 and the protein complex was further identified through Pearson correlation analysis. The motor proteins RHOT, RHOT2, TRAK1, and TRAK2 were positively associated with TRPM7 (Supplementary Fig. 14e).

These results suggested that TRPM7 may play a role in regulating intercellular mitochondrial communication. To verify this hypothesis, SLCs were treated with a shTRPM7-expressing lentivirus. Western blotting revealed that TRPM7 protein levels were markedly downregulated (Fig. 5B, C). Knockdown of TRPM7 in SLCs showed significantly lower efficiency in transferring mitochondria from donors to acceptors (Fig. 5D, E). Others have reported that TRPM7 is activated when stimulated by ROS[35], and we found increased mitochondrial superoxide in resident macrophages 24 h after testicular torsion (Supplementary Fig. 15a, b) and decreased mitochondrial membrane potential (Supplementary Fig. 15c, d) compared to the control group. We further elucidated whether TRPM7 regulation of mitochondrial

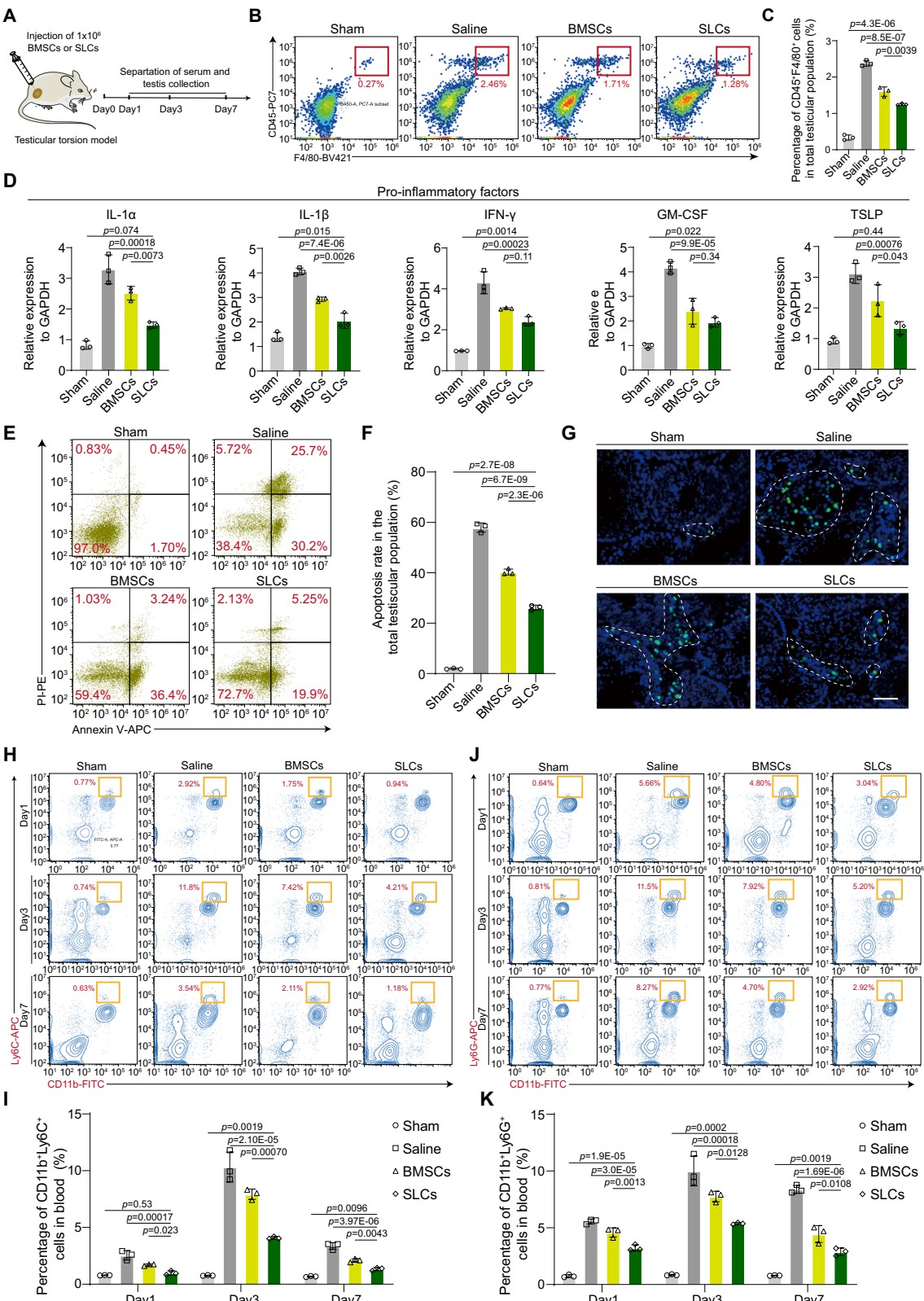

transfer is stimulated by mitochondrial superoxide. Using N-acetylcysteine (NAC) (a scavenger of ROS) and $H_2O_2$ (a source of ROS) (Supplementary Fig. 15e), we found that the interception of ROS released by macrophages significantly reduced the rate of mitochondria that underwent transfer, which was reversed when additional $H_2O_2$ was administered (Supplementary Fig. 15f, g). We further performed additional experiments using mito-TEMPO (a mitochondrial

superoxide inhibitor) to clarify the mitochondrial superoxide that is released to activate TRPM7. Flow cytometry revealed that mito-TEMPO treatment significantly reduced the rate of mitochondria that underwent transfer (Supplementary Fig. 15h, i). These results indicated that mitochondrial superoxide influences mitochondrial transfer from SLCs to macrophages. Furthermore, knockdown of TRPM7 resulted in an impaired transfer rate when treated with $H_2O_2$ (Fig. 5F, G), and the

**Fig. 3 | Transplanted SLCs attenuate inflammatory reactions. A** Schematic of the experimental procedure for detecting the percentage of macrophages in total testicular cells at different time points. **B** Flow cytometry for detecting the percentage of CD45⁺F4/80⁺ macrophages among total testicular cells on day 3. **C** Quantitative analysis of the percentage of CD45⁺F4/80⁺ macrophages on day 3. Data are presented as the means ± SDs, *n* = 3 biological replicates for each group, one-way ANOVA. **D** RT-qPCR revealed the expression of inflammatory cytokines in CD45⁺F4/80⁺ macrophages sorted from the testes 3 days after cell transplantation. Data are presented as the means ± SDs, *n* = 3 biological replicates for each group, one-way ANOVA. **E** Flow cytometry for detecting the percentage of apoptotic cells. **F** Flow cytometry-based quantification of apoptotic cells. Data are presented as the means ± SDs, *n* = 3 biological replicates for each group. **G** Representative image showing apoptotic cells in the testes. Scale bar, 50 μm. **H** Representative flow cytometry profiles showing the percentage of CD11b⁺Ly6C⁺ cells in the blood of each group at 1, 3, and 7 days after cell transplantation. **I** Quantitative analysis of the percentage of CD11b⁺Ly6C⁺ cells in the sham saline, BMSCs and SLCs groups at 1, 3, and 7 days. Data are presented as the means ± SDs, *n* = 3 biological repeats for each group, one-way ANOVA. **J** Representative flow cytometry profiles showing the percentage of CD11b⁺Ly6G⁺ cells in the blood of each group at 1, 3, and 7 days after cell transplantation. **K** Quantification of the percentage of CD11b⁺Ly6G⁺ cells in the sham, saline, BMSCs and SLCs groups at 1, 3, and 7 days. Data are presented as the means ± SDs. *n* = 3 biological replicates for each group, one-way ANOVA was used. Source data are provided as a Source Data file.

results suggested that mitochondrial superoxide elicits mitochondrial transfer in a manner partly dependent on TRPM7.

TRPM7 has been identified as a key regulator of the cytoskeleton that operates via downstream modulation of myosin II[36], and myosin II has been observed in donor-acceptor connections through nanotubes[37]. We then further assessed whether myosin II was present in the SLCs-macrophage connections, while blebbistatin (which slows the phosphate release of myosin II) treatment induced an increase in the mitochondrial transfer rate (Fig. 5H, I). By immunofluorescence staining, we found myosin II inside the tunneling nanotube (Fig. 5J). These observations strongly support that TRPM7 senses mitochondrial superoxide and promotes the transfer of mitochondria by modulating myosin II.

We next ascertained whether suppressing TRPM7 expression directly affects the rescue of testicular torsion by SLCs transplantation in mice. SLCs were transfected with the shTRPM7 lentivirus, and then SLCs and SLCs^shTRPM7 transfected with the Mito-DsRed lentivirus were transplanted into testes after testicular torsion. Similar to the aforementioned results, flow cytometry analysis revealed that Mito-DsRed levels in F4/80⁺CD45⁺ macrophages were higher in the SLCs group than in the SLCs ^shTRPM7 group after 24 h (Fig. 5K, L). Knockdown of TRPM7 did not exert any effects on the differentiation of SLCs (Supplementary Fig. 16a–c). On the 28th day, the testicular weights of the whole body were significantly lower in the SLCs^shTRPM7 group than in the SLCs group (Fig. 5M–O). Histopathological imaging showed that the germ cell layer was significantly reduced in the SLCs^shTRPM7 group compared with the SLCs group (Fig. 5M–O). Furthermore, by staining with PNA, we found that the number of PNA⁺ cells significantly decreased in the SLCs^shTRPM7 group compared to the SLCs group (Fig. 5P, Q). Based on all the above results, we concluded that TRPM7 enhances the capacity of SLCs to repair tissue through mitochondrial transfer. Additionally, we examined the endogenous SLCs conditional knockdown (cKD) of TRPM7 and whether they exhibit any defects in spermatogenesis or tissue homeostasis after testicular torsion. We generated AAV8 vectors that carried shTRPM7 (AAV8-shTRPM7), and 8- to 10-week-old mice were interstitially injected with AAV8-shTRPM7 at doses of 8 × 10^10 gc/testis[38] (Supplementary Fig. 17a). Histological analysis of the testes 7 days after vector injection showed the coexpression of mCherry and the SLCs marker nestin (Supplementary Fig. 17b). After testicular torsion, we found that the ratio of CD45⁺F4/80⁺ macrophages was significantly increased in the AAV8-shTRPM7-treated group compared to the control group (Supplementary Fig. 17c d), while the number of germ cells was markedly reduced on day 7 after testicular torsion (Supplementary Fig. 17e, f). These results suggested that TRPM7 expressed on endogenous SLCs played an important role in testis tissue homeostasis after testicular torsion.

### SLCs exert testicular anti-aging effects partly through TRPM7-mediated mitochondrial transfer from transplanted cells to macrophages

Aging is characterized by a chronic, persistent and low-grade inflammatory response[39], and mitochondrial dysfunction plays an important role in inflammation in aging testes[40,41]. Proteomic sequencing analysis of testicular mesenchymal exosomes showed differential expression profiles in young and aging testes (Fig. 6A and Supplementary Data 1). GO analysis of aging and inflammation-related genes suggested that the upregulated genes in aging testes were enriched in "positive regulation of leukocyte activation", "regulation of response to oxidative stress", "cellular senescence", and "activation of immune response" (Fig. 6B). It is worth noting that genes annotated to "protein-containing complex assembly", "microtubule-based transport", "transport along microtubules" and "regulation of cell communication" were downregulated in aging testes compared to young testes (Fig. 6C). Moreover, we found that the macrophage mitochondrial superoxide levels were significantly increased (Supplementary Fig. 18a, b) and that membrane potential was decreased in the testes of 24-month-old mice compared to 3-month-old mice (Supplementary Fig. 18c, d). Based on the above results, we hypothesized that TRPM7-mediated mitochondrial transfer from SLCs to macrophages may similarly occur in aging testes. To confirm this, SLCs or SLCs^shTRPM7 transfected with the Mito-DsRed lentivirus were transplanted into the testes of 24-month-old mice. After 24 h, flow cytometry analysis revealed that the Mito-DsRed levels in macrophages were higher in the SLCs group than in the SLCs^shTRPM7 group (Fig. 6D, E). In addition, recipient macrophages exhibited lower superoxide levels (Supplementary Fig. 18e, f) and higher membrane potential (Supplementary Fig. 18g, h) in the SLCs group than in the SLCs^shTRPM7 group and saline groups. In line with our hypothesis, these results demonstrated that in aging testes, SLCs can transfer mitochondria to macrophages in a TRPM7-dependent manner.

To further assess the effect of suppression of TRPM7 expression on the therapeutic effect in aging testes, we detected endocrine function and spermatogenesis in the testes. On the 28th day, the serum testosterone levels were increased in the SLCs group compared to the SLCs^shTRPM7 group (Supplementary Fig. 19a). Flow cytometry indicated that the abundance of LCs was decreased in the SLCs^shTRPM7 group compared to the SLCs group (Supplementary Fig. 19b, c), and there was no significant change in bone density or muscle fiber area (Supplementary Fig. 19d–f). Histopathological imaging showed that the thickness of the seminiferous tubules and germ cells in the seminiferous epithelium was significantly decreased (Fig. 6F, G). Moreover, the proportions of VASA⁺ and SYCP3⁺ cells were significantly decreased in the SLCs^shTRPM7 group compared to the SLCs group (Fig. 6H–J). Accordingly, there were significantly fewer PNA⁺ cells in the SLCs^shTRPM7 group than in the SLCs group (Fig. 6K, L). Semen analysis showed that the epididymal sperm count and sperm motility were significantly lower in the SLCs^shTRPM7 group (Fig. 6M, N). Histological analysis showed the presence of massive sperm in the cauda epididymis in the SLCs group (Fig. 6O). In the in vitro fertilization assay, most sperm were fertilized and developed into 2-cell embryos, whereas the sperm from the SLCs^shTRPM7 group exhibited markedly lower fertilization rates (79.69% in the SLCs^shTRPM7 group versus 85.52% in the SLCs group). Knockdown of TRPM7 further decreased the morula and blastocyst formation rates (90.20% in the SLCs^shTRPM7 group versus 95.38% in the

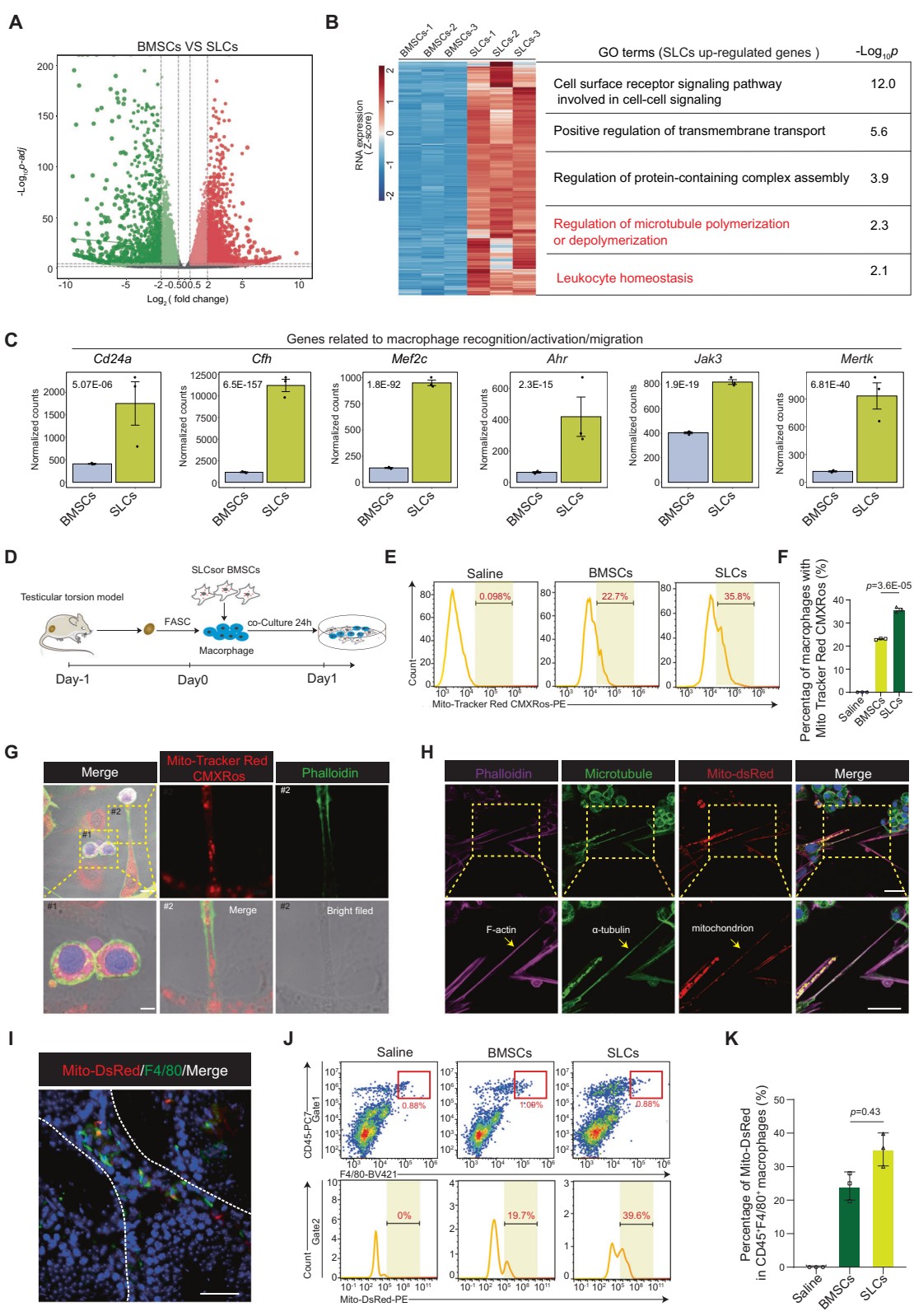

SLC group) (Fig. 6P). To detect mouse fertility, a male mouse was mated with 3-month-old female mice at a ratio of 1:2 (Fig. 6Q). The number of pups per litter within 4 months in the SLCs^shTRPM7 groups lower than in the SLCs group (Fig. 6R, S), suggesting that suppressing TRPM7 expression weakened the fertility of aging mice. We further examined the effects of TRPM7 knockdown on fertility status in aging mice at 3 months. Equal amounts of SLCs^zsGreen and SLCs^zsGreen/shTRPM7

were transplanted into the testicular interstitium (Supplementary Fig. 20a). After 3 months, flow cytometry analysis revealed that the ratio of SLCs^zsGreen was significantly higher than that of SLCs^zsGreen/shTRPM7 (Supplementary Fig. 20b, c). In addition, sperm counts and sperm motility in the epididymis were notably higher in the SLCs^zsGreen group than in the SLCs^zsGreen/shTRPM7 group (Supplementary Fig. 20d–f). In vitro fertilization (IVF) further confirmed the improved sperm quality, and

**Fig. 4 | Nanotubes mediate mitochondrial transfer from SLCs to macrophages.** **A** Volcano plot showing the DEGs between BMSCs and SLCs. The Wilcoxon test was used to calculate p values and fold changes. **B** Representative GO terms for the genes upregulated in SLCs versus BMSCs. The statistical test is a hypergeometric test. **C** Bar graphs showing the expression levels of macrophage homeostasis-related genes (BMSCs, $n = 3$ biologically independent cells, SLCs, $n = 3$ biologically independent cells). The statistical test is a hypergeometric test. **D** Schematic illustration of the experimental workflow for coculture experiments. **E** Flow cytometry analysis of mitochondrial transfer to total macrophages after 24 h. Macrophages labeled with CellTrace Violet were cocultured with BMSCs or SLCs labeled with MitoTracker Red CMXRos. **F** The percentage of mitochondria transferred to macrophages in each group was analyzed and graphed. Data are presented as the means ± SDs, $n = 3$ biological replicates for each group, unpaired two-tailed Student's t test. **G** Representative confocal microscopy image of macrophages stained with CellTrace Violet cocultured with SLCs labeled with MitoTracker Red CMXRos. The nanotube was stained with phalloidin. Scale bar, 10 µm; the larger scale bar, 2 µm. **H** Representative confocal microscopy image of macrophages cocultured with SLCs labeled with Mito-dsRed. The nanotube was stained with phalloidin and microtube. The yellow arrow indicates the nanotube. Scale bar, 10 µm; large scale bar, 5 µm. **I** Representative confocal microscopy image of SLCs^Mito-DsRed mitochondrial transfer to F4/80+ (green) macrophages. Scale bar, 50 µm. **J** Flow cytometry analysis of mitochondrial transfer from SLCs^Mito-DsRed or BMSCs^Mito-DsRed to CD45+F/480+ macrophages cells. **K** Quantitative analysis of the percentage of mitochondrial transfer rate in vivo. Data are presented as the means ± SDs. $n = 3$ biological replicates for each group, unpaired two-tailed Student's t test. Source data are provided as a Source Data file.

we found significantly higher rates of 2-cell embryos and blastocysts in the SLCs^zsGreen group (Supplementary Fig. 20g–i). These findings suggested that knockdown of TRPM7 impairs the therapeutic effect of SLCs to promote fertility recovery at 3 months in aging mice.

Together, our studies demonstrated that SLCs can actively connect to activated macrophages to share mitochondria as a strategy to attenuate inflammatory reactions (Fig. 7). Our findings prove that this mechanism is conserved over different acute and chronic inflammation model systems.

## Discussion

SLCs transplantation has been reported to significantly restore testosterone production and improve spermatogenesis in aged or LCs-disrupted animal models. Although the steroidogenic lineage differentiation potential is undisputable, our study revisits the hypothesis that multipotent stromal cells maintain tissue homeostasis. We focused on the significance of the immunomodulatory capacity of SLCs in stem cell-based therapy. The regulatory role of SLCs in testis homeostasis via the transfer of mitochondria to activated macrophages during acute and chronic inflammatory responses was explored. Macrophages acquire mitochondria under the regulation of TRPM7. This process triggers the rapid shift of macrophages from a proinflammatory phenotype toward an anti-inflammatory phenotype, and SLCs reshape the local immune microenvironment to protect the local microenvironment from acute and chronic immune challenge. Our results provide new insight into the mechanism by which SLCs therapies restore testis function, thereby treating reproductive system diseases.

Some studies have documented the involvement of resident multipotent stromal cells driving tissue regeneration in wound healing and inflammation by directly affecting immune cells[42]. SLCs are multipotent stromal cells located in the interstitial testis and display multilineage differentiation capacity during homeostasis. In this study, SLC-based cell transplantation strategies did not achieve this promise, shifting attention to their immunomodulatory activity. Macrophages contribute to the maintenance of immune homeostasis in the testicular interstitium, which constitutes a unique isolated physiological environment needed for germ cell development[43]. Testicular torsion is a urological acute injury caused by tissue ischemia–reperfusion injury following acute blood flow reperfusion to the twisted spermatic vessel[44]. We have already observed that the transplanted SLCs^zsGreen gradually decreased during the early time, and very few cells were found on day 7 after testicular torsion (approximately 0.3%). Most transplanted cells had disappeared, which limits the number of transplanted SLCs differentiated into LCs. In line with a previous study, mesenchymal stem cells display poor survival and engraftment rates in vivo, and their short lifespan could be explained by the stresses encountered after local transplantation, such as nutrient deprivation and proinflammatory cytokines[45,46]. Moreover, a previous study reported that in vivo differentiated stem cells elicit enhanced immunogenicity, thereby increasing their chance for rejection[47,48]. The state of SLC differentiation may affect their immunogenicity, leading to only a small number of exogenous LCs surviving in the testis. In our study, we found that transplanted SLCs had the ability to protect ischemic testicular cells from death, which could protect endogenous SLCs from apoptosis. A previous study reported that inflammatory cytokines, including interleukin 6 (IL-6), suppress the differentiation of SLCs, causing a decrease in testosterone production. Other studies have also shown that coculture with aging macrophages, in which the expression of most cytokine genes is upregulated, results in a significant reduction in the proliferation of SLCs[49]. We found that although most transplanted cells had disappeared (Fig. 2G, H), the surviving cells attenuated macrophage inflammatory responses, which promoted the subsequent recruitment of leukocytes to the affected testis. An influx of neutrophils and monocytes into the testis occurs, and this phenomenon is gradually resolved following SLC transplantation, which creates a regenerative immune microenvironment for endogenous SLC regeneration. In addition to cellular differentiation and close interactions with macrophages, other mechanisms related to paracrine factors may be involved in the therapeutic effects of endogenous SLCs to expand and replenish LCs after testicular torsion. In summary, we revealed a new mechanism, apart from differentiation into LCs, by which SLCs remodel testicular function to treat reproductive system inflammatory disease.

During ischemia–reperfusion, which occurs in several diseases, including stroke, myocardial infarction, and organ transplants, mitochondria are primary targets and the origin of tissue damage[50]. Against this background, the current study further explored the involvement of mitochondria-related apoptosis and cell cycle machinery systems in testicular torsion using a rat model[51]. As the powerhouses of immunity, mitochondria are critical regulators of immune cell activation, differentiation, and survival. Dysfunctional mitochondria exhibit impaired cell membrane integrity, produce large amounts of reactive oxygen species, and release mitochondrial DNA, which directly regulates the immune response[52]. Compared to mitochondrial biogenesis, mitochondrial transfer has emerged as a highly effective therapeutic approach for replenishing the bioenergetic requirements of impaired cells. For example, human CD34+ hematopoietic progenitor cells (HPCs) can also acquire mitochondria from BMSCs in response to acute bacterial infection, which causes bioenergetic changes that underpin emergency granulopoiesis[53]. Moreover, hijacking mitochondrial transfer from immune cells metabolically empowers cancer cells to develop new immune evasion strategies[54]. These findings demonstrate the role of mitochondria as a signal transduction pathway that allows cells to adapt to environmental demands. Here, we found that mitochondrial ROS were increased in resident macrophages following testicular torsion, and this was accompanied by decreased mitochondrial membrane potential, which caused macrophages to secrete inflammatory factors. To resolve inflammation and restore tissue homeostasis, macrophages can acquire anti-inflammatory properties to promote tissue repair. The mechanism by which infused SLCs restore testis homeostatic functions through their effects

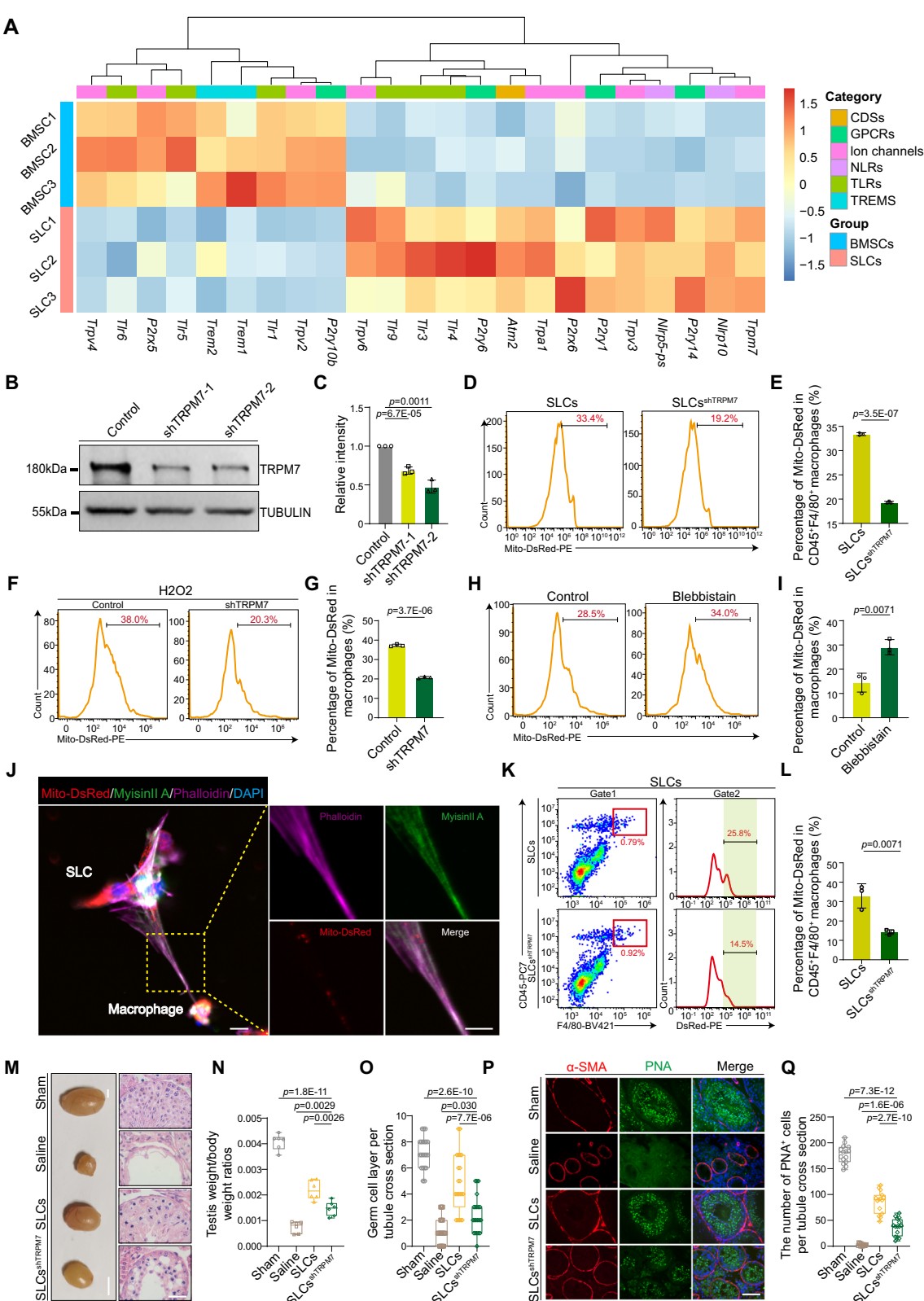

on macrophages is the focus of inquiry. A key finding was the detection of nanotube-like structures that physically connect SLCs and macrophages. SLCs exhibited a higher mitochondrial transfer rate to macrophages and an increased capacity to attenuate testicular torsion-induced inflammation at the early stage. Previous reports supporting the donation of mitochondria have corroborated these findings, demonstrating the mitigation of inflammatory responses in

macrophages. Other studies have demonstrated that resident macrophages are involved in intercellular mitochondrial transfer in various tissues[55], which highlights a new critical function of macrophages in tissue homeostasis. Collectively, these findings have important implications. First, we reveal that SLCs regulate the immune microenvironment in the testis via mitochondria transfer to resident macrophages, a transcellular process that has never been reported for

**Fig. 5 | TRPM7 is needed for mitochondrial transfer via TNT. A** Heatmap showing the expression of pattern recognition receptors on BMSCs and SLCs. **B** Western blot analysis of TRPM7. **C** Quantitative analysis of TRPM7. Data are presented as the means ± SDs, $n = 3$ biological replicates for each group, one-way ANOVA. **D** Flow cytometry analysis of the mitochondrial transfer rate. **E** Quantitative analysis of the mitochondrial transfer rate. Data are presented as the means ± SDs, $n = 3$ biological replicates for each group, unpaired two-tailed Student's t test. **F** The percentage of mitochondria transferred to macrophages treated with $H_2O_2$. **G** Quantitative analysis of the mitochondrial transfer rate. Data are presented as the means ± SDs. $n = 3$ biological replicates for each group, unpaired two-tailed Student's t test. **H** The percentage of mitochondria transferred to macrophages after treatment with blebbistatin. **I** Quantitative analysis of the mitochondrial transfer rate. Data are presented as the means ± SDs, $n = 3$ biological replicates for each group, unpaired two-tailed Student's t test. **J** Macrophages cocultured with SLCs[Mito-DsRed]. Scale bar,

10 μm; large scale bar, 5 μm. **K** Knockdown of TRPM7 reduced mitochondrial transfer rate. **L** Quantitative analysis of the mitochondrial transfer rate in vivo. Data are presented as the means ± SDs. $n = 3$ biological replicates for each group, unpaired two-tailed t test. **M** Bright field diagram of testicular (scale bar, 2 mm) and H&E staining of testis samples (scale bar, 50 μm). **N, O** Quantification of the testis weight of the whole body and germ cell layers, $n = 6$ biological replicates for each group. Data are presented as the means ± SDs, one-way ANOVA. **P** Immunostaining of α-SMA (red) and PNA (green). Scale bar, 50 μm. **Q** Quantitative analysis of the PNA[+] cells. $n = 6$ biological replicates for each group. Data are presented as the means ± SDs, one-way ANOVA. All box-and-whisker plots denote the maximum (top whisker), 75th (top edge of box), 25th (bottom edge of box), and minimum (bottom whisker) percentiles and the median (line in box). Source data are provided as a Source Data file.

the male reproductive system. Second, failure of this mechanism may lead to pathological conditions, and thus, targeting mitochondria may be a promising method to restore testis function in the context of immune imbalance.

The crosstalk between MSCs and the inflammatory microenvironment is essential for the reparative process and homeostatic maintenance[56]. MSCs can actively respond to damage signals in tissues under dynamic pathological conditions, and they can mobilize various components in the tissue immune microenvironment to promote tissue repair[57]. Numerous endogenous molecules and mitochondrial respiration byproducts, such as ROS released by damaged cells, can be recognized by PRRs[58]. Intriguingly, TRPM7 is one of the most abundant PRRs that is detected on SLCs. It is a ubiquitously expressed nonselective cationic ion channel with intrinsic serine/threonine kinase activity. Studies have shown that it is sensitive to multiple signals related to inflammation, such as caspase-mediated cleavage and extracellular pH. Here, we show that SLCs can sense excessive ROS via TRPM7 and quickly transfer mitochondria to macrophages to inhibit the onset of the emergency inflammatory response. Previous studies have shown that TRPM7 senses oxidative stress to stimulate the release of $Zn^{2+}$ from intracellular vesicles, which supports the notion that SLCs sense damage signals, including ROS. Several reports suggest that TRPM7 positively regulates actin cytoskeletal remodeling by phosphorylating the myosin IIA heavy chain. These cytoskeletal remodeling processes contribute to the formation of TNTs[59]. Our results demonstrated that TRPM7 was positively associated with the RHOT1, RHOT2, TRAK1, and TRAK2 genes, all of which have been reported to promote mitochondrial transfer through the nanotube. Suppression of TRPM7 by short interfering RNA reduced intercellular mitochondria transfer. Using blebbistatin as a selective inhibitor of myosin II[60], we found a significant increase in mitochondrial transfer between cells, providing evidence for the involvement of myosin II in the aforementioned transfer process. Collectively, the present results demonstrate that TRPM7 plays a crucial role in the formation of a functional rescue network. By gaining a deeper understanding of and regulating the plasticity of SLCs in immunoregulation, we can advance these strategies to a greater degree. However, there are many potential limitations that need to be investigated in future research. We did not investigate in detail the cellular mechanism underlying the regulatory functions of TRPM7 on the contact formation and fusion of membranes. Moreover, whether other factors contribute to mitochondrial transfer via TRPM7-independent mechanisms was not investigated.

In conclusion, our work uncovers a crucial role of SLCs in restoring testis function from immune-related disorders in models such as testicular torsion and aging. SLCs can quickly and efficiently sense damage signals and establish connections with resident macrophages, thereby inhibiting several inflammatory responses. This process creates a favorable environment that improves fertility in male animals. The alternative mechanism that serves as a theoretical basis for SLC-based cell therapy or possible cell-free therapeutic approaches

is especially significant in the treatment of immune-associated infertility through clinical applications.

## Methods

### Ethics statement

All animal experiments described in this study were approved by the Institute of Animal Care and Use Committee of the First Affiliated Hospital of Sun Yat-sen University (no. 2021000022). All animal experiments abided by the ARRIVE guidelines.

### Animals

C57BL/6 J mice (male, 8-10 weeks old) were purchased from the Animal Center at the Medical Laboratory of Guangdong Province. Actin-zsGreen mice were purchased from GuangDong GemPharmatech Co., Ltd. All experimental procedures involving animals were in accordance with the guidelines of the Animal Care and Use Committee of the First Affiliated Hospital of Sun Yat-sen University (Guangzhou, China). All mice were housed in a pathogen-free animal facility with 50% humidity, 20 °C temperature, and a 14-h light/10-h dark cycle with ad libitum access to water and food. Testicular ischemia–reperfusion models were established as follows[61]: mice were anesthetized with sodium pentobarbital, and the right testis was rotated 720° in a clockwise direction and fixed to the scrotum with silk suture for 1 h, after which the left testis was removed. Detection was performed by untwisting the testis. The testes that returned to red from dark purple 5 min after detorsion were used in subsequent studies. Approximately $1 \times 10^6$ BMSCs or SLCs in 20 μL of saline were injected into the interstitium of the recipient testes immediately after detorsion ($n = 6$ mice per group), whereas control animals ($n = 6$ mice per group) received the same volume of saline. After surgery, the animals were kept in the original environment. At the indicated time points after surgery, animals in each group were euthanized by an intraperitoneal injection with 150 mg/kg pentobarbital sodium solution. Testes and serum from all animals were isolated and tested for further analysis.

### Cell isolation and culture

Isolation and culture of primary mouse SLCs were performed as follows: the testes were dissected from 7-day-old C57BL/6 mice, and the tunica albuginea was carefully removed, after which the testes were minced into small pieces. Interstitial cells were dissociated from seminiferous tubules using 1 mg/mL collagenase type IV (Gibco, 17104-019) in Dulbecco's modified Eagle's medium (DMEM)/F12 (Gibco, 11320033) at 37 °C for 15 min. After the addition of DMEM/F12 containing 10% fetal bovine serum (VISTECH, SE100-011) to stop collagenase activity, the samples were centrifuged at $1500 \times g$ for 3 min at room temperature. The pellets were resuspended in PBS and filtered through a 70 μm filter. The CD51[+] cells were enriched by FACS using an Influx Cell Sorter (Becton Dickinson), after which they were seeded in SLCs culture medium. The medium consisted of DMEM/F12 supplemented with 1 nM dexamethasone (Sigma, D4902), 1 ng/mL

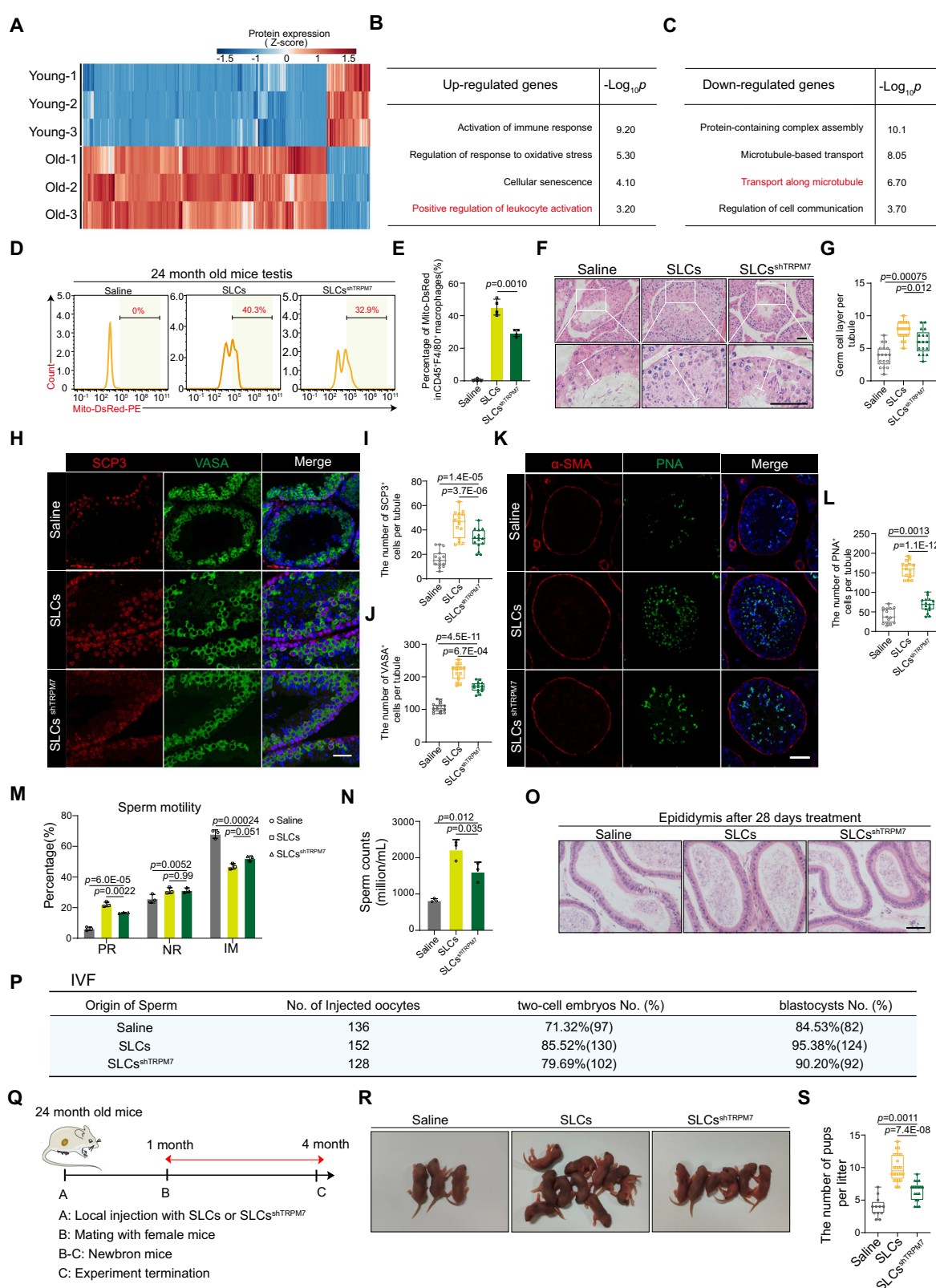

LIF (Sigma, LIF2005), 5 µg/L insulin-transferrin-sodium selenite (Gibco, 41400045), 5% chicken embryo extract (US Biologicals, C3999), 0.1 mM β-mercaptoethanol (Gibco, 21985023), 1% nonessential amino acids (Gibco, 11140050), 1% N2 (Gibco, 17502001) and 2% B27 (Gibco, 17504044) supplements, 20 ng/mL basic fibroblast growth factor (Invitrogen, RFGFB50), epidermal growth factor (PeproTech, 31509), platelet-derived growth factor (PeproTech, 31518), and oncostatin M

(PeproTech, 30010). To isolate BMSCs, 7-day-old C57BL/6 mice were sacrificed, and both sides of their femurs and tibias were removed and temporarily stored in DMEM (GIBCO, C11885500BT) containing 1× penicillin/streptomycin (Gibco, 15140122) on ice. Subsequently, the majority of the muscles and connective tissues were scraped, and the ends of the femurs and tibias were cut using scissors to expose the bone marrow. The bone marrow was sucked using a syringe and

**Fig. 6 | Therapeutic effects of SLCs versus SLCs^shTRPM7 on testicular function in aging mice. A** Heatmap of DEGs in exosomes extracted from testis interstitial fluid. The Wilcoxon test was used to calculate p values and fold changes.
**B, C** Representative GO terms for the upregulated and downregulated genes. The statistical test is a hypergeometric test. **D** Flow cytometry analysis of mitochondrial transfer from SLCs^Mito-DsRed or BMSCs^Mito-DsRed to CD45⁺F/480⁺ macrophages.
**E** Quantitative analysis of mitochondrial transfer rate in vivo. Data are presented as the means ± SDs. *n* = 3 biological replicates for each group, one-way ANOVA. **F** H&E staining of testis samples (scale bar, 50 μm). **G** The germ cell layers were evaluated. *n* = 6 biological replicates for each group. Data are presented as the means ± SDs, one-way ANOVA. **H** Immunostaining of VASA (green) and SCP3 (red).
**I, J** Quantitative analysis of the VASA⁺ or SCP3⁺ cells. Scale bars, 50 μm. Data are presented as the means ± SDs. *n* = 3 biological replicates for each group, one-way ANOVA. **K** Immunostaining of α-SMA (red) and PNA (green). Scale bar, 50 μm.

**L** Quantitative analysis of the number of PNA⁺ cells. Data are presented as the means ± SDs, *n* = 3 biological replicates for each group, one-way ANOVA.
**M, N** Quantitative analysis of the sperm motility and sperm counts. PR progressive motile, NP nonprogressive motile, IM immotility. Data are presented as the means ± SDs. *n* = 3 biological replicates for each group, one-way ANOVA.
**O** Histological analysis of cauda epididymides. Scale bars, 50 μm. **P** The trilinear table shows all embryo injection, two-cell embryo and blastocyst data. **Q** Timeline of the mating assessment. **R** Offspring from the saline, SLCs and SLCs^shTRPM7 groups. **S** Quantitative analysis of the number of pups per litter. Data are presented as the means ± SDs. *n* = 6 biological replicates for each group, one-way ANOVA. All box-and-whisker plots denote the maximum (top whisker), 75th (top edge of box), 25th (bottom edge of box), and minimum (bottom whisker) percentiles, and the median (line in box). Source data are provided as a Source Data file.

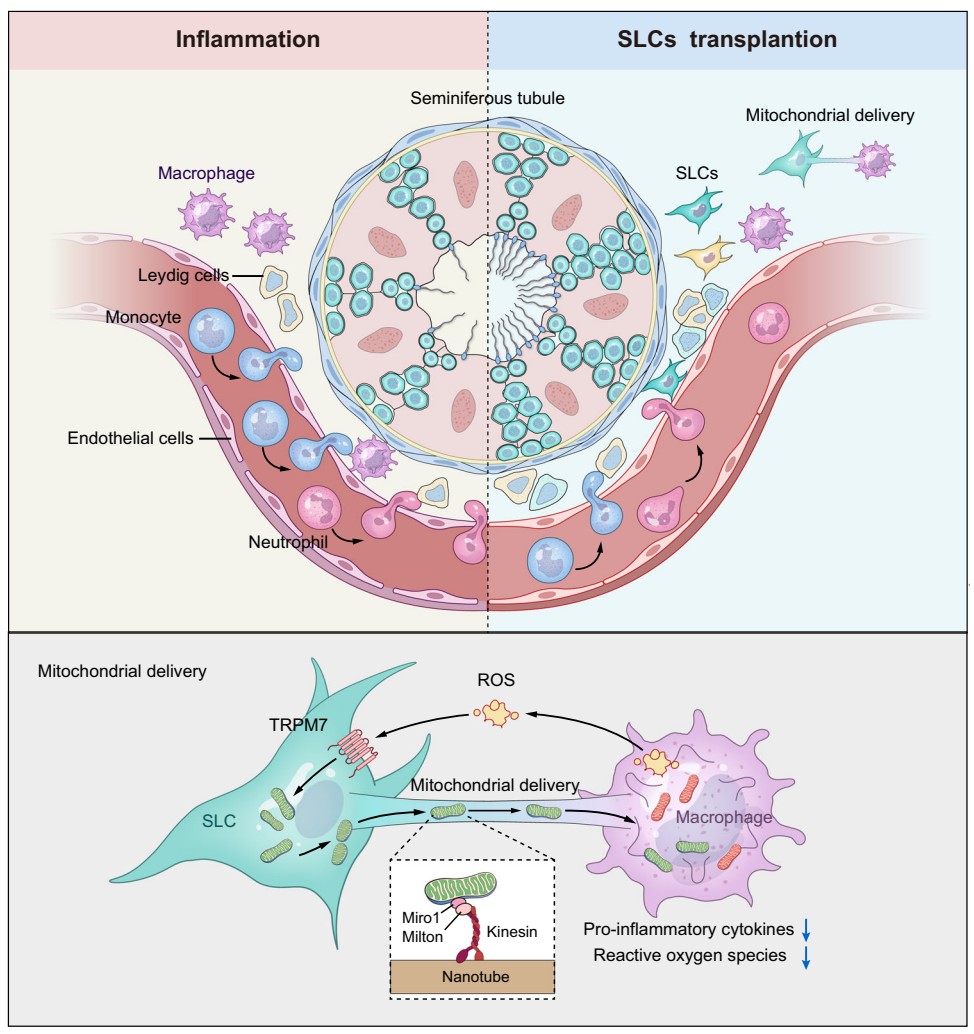

**Fig. 7 | Schematic illustration of SLCs transfer mitochondria to testis tissue-resident macrophage.** ROS released from activated macrophages inducing mitochondrial transfer from SLCs to macrophages in a TRPM7-mediated manner, which protected the macrophages from dysfunction, suppressing secretion of pro- inflammatory factors and reshaped the local immune micro-environment. Furtherly, SLCs accelerated the recovery of endogenous Leydig cells, effectively prevented spermatogenesis and fertility.

flushed into fresh medium. A single-cell suspension was obtained by filtering the above flush fluid through a 70 μm cell strainer, followed by centrifugation and resuspension. The cell suspension was transferred into a T25 culture flask and cultured in DMEM supplemented with 10% FBS and 1 × penicillin/streptomycin. The purified BMSCs were obtained by gradually replacing the medium with fresh medium to remove nonadherent cells at different time points and subculturing the cells for 3 weeks. Subsequently, the cells were stained with specific

antibodies to assess the expression of stem cell protein markers. Both SLCs and BMSCs were cultured at 37 °C in a 5% CO2 water-jacketed incubator. The cells were propagated every 3 days, and cells from similar passages were used in all assays.

### Coculture of macrophages with BMSCs or SLCs
The SLCs or BMSCs were labeled with MitoTracker Red CMXRos (Invitrogen, M7512), washed twice and incubated in medium for 1 h to

remove excess unbound MitoTracker dyes. Primary macrophages were isolated from mouse testes by flow cytometry sorting and labeled with CellTrace Violet (Invitrogen, C34571). Cocultures were established by incubating macrophages and BMSCs or SLCs in a 1:1 ratio in their respective medium for 24 h.

## Fluorescence microscopy

Cells grown on a dish were fixed in 4% paraformaldehyde (Beyotime, P0099) at room temperature for 30 min, washed three times (15 min each time) using PBS, permeabilized by incubation with 0.1% Triton X-100 (Beyotime, P0099) at 4 °C for 10 min and washed three times using PBS. F-actin localization was determined by staining using phalloidin-Alexa 488 (Invitrogen, A12379) at room temperature for 1 h. For nuclear staining, cells were incubated with DAPI (Thermo Scientific, 62248) for 10 min. The testicular samples were harvested, immediately fixed in paraformaldehyde and embedded in paraffin after gradient alcohol dehydration. After deparaffinization and antigen retrieval, the testicular tissues were incubated with primary antibodies overnight at 4 °C. The sections were washed three times using PBS and incubated with secondary antibodies for 1 h at room temperature the next day. Then, the sections were washed three times using PBS and incubated with secondary antibody and/or peanut agglutinin (PNA, Sigma, L7381) for 1 h at room temperature. The nuclei were stained with DAPI for 10 min. The primary antibodies in this assay were as follows: mouse anti-DDX4 (Abcam, ab27591, 1:500), rabbit anti-SYCP3 (Abcam, ab15093, 1:400), rabbit anti-α-SMA (Abcam, ab5694, 1:500), rat anti-F4/80 (Abcam, ab6640, 1:200), mouse anti-3β-HSD (Santa Cruz Biotechnology, 1:100), rabbit anti-CYP11A1 (Cell Signaling, 14217, 1:400), rabbit anti-Myosin IIa (Cell Signaling, 49349, 1:400), rabbit anti-TRPM7 (Affinity, DF7513, 1:400), and mouse anti-Nestin (Abcam, ab134107, 1:400). The secondary antibodies used in this assay were as follows: goat anti-mouse Alexa Fluor 488 (Invitrogen, A28175, 1:400), goat anti-rabbit Alexa Fluor 488 (Abcam, ab150077, 1:400), goat anti-rabbit Alexa Fluor 555 (Abcam, ab150078, 1:400), and goat anti-rat Alexa Fluor 488 (Abcam, ab150165, 1:400). TUNEL staining was performed using a commercial in situ apoptosis detection kit (Roche, 11684795910), as instructed by the manufacturer. Confocal images were acquired using a Zeiss LSM 800 microscope with ZEN 2.3 software.

## Reverse transcription and real-time qPCR

Total RNA was purified from the testis or cultured cells using TRIzol reagent (Fisher Scientific, 15596018), as instructed by the manufacturer. cDNA was synthesized using murine leukemia virus reverse transcriptase and oligo-dT primers (Fermentas). Quantitative PCR (qPCR) was performed using 2×PCR Master Mix (GenStar, A301-10), as instructed by the manufacturer. Signals were detected using a Light Cycler 480 Detection System (Roche). The primer sequences are listed in Supplementary Table S1.

## Flow cytometry

Testicular cells derived from C57BL/6 mice at postnatal day 7 or 8-10 weeks were minced, incubated with 1 mg/mL collagenase type IV at 37 °C for 15 min, centrifuged at 1500 × g for 5 min at room temperature, and filtered using staining buffer (PBS with 0.5% BSA) through a 70 µm cell strainer to obtain a single cell suspension. Depending on the experimental design, fixation and permeabilization were performed as stated above. Blood samples were resuspended in 10 mL of red blood cell lysis buffer and incubated at 4 °C for 10 min. Then, the samples were centrifuged at 1200 × g for 5 min at room temperature and filtered using staining buffer (PBS with 0.5% BSA) through a 70 µm cell strainer to obtain a single cell suspension. The cells were stained with antibodies in the dark on ice at room temperature. The antibodies that were used were as follows: CD51-BV421 antibody (BD Bioscience, 740062, 1:100), F4/80-BV421 antibody (BioLegend, 123137, 1:100),

CD45-PE/CY7 antibody (BioLegend, 103113, 1:100), CD11b-FITC antibody (BioLegend, 101205, 1:100), Ly6C-APC antibody (BioLegend, 128015, 1:100), Ly6G-APC antibody (BioLegend, 127613, 1:100), CD86-FITC antibody (BioLegend, 105109, 1:100), CD206-FITC antibody (BioLegend, 141703, 1:100), Agr1-PE antibody (R&D Systems, IC5868P, 1:100), INOS-PE antibody (BioLegend, 128015, 1:100), anti-LHR (Alomone labs, ALR-010, 1:100), and donkey anti-rabbit IgG-PE (BioLegend, 406421, 1:100). Apoptotic levels in each group were determined using the Annexin V-APC/PI double-dye cell apoptosis detection kit (Elabscience, E-CK-A217) according to the manufacturer's instructions. Briefly, the obtained cells were washed twice using PBS and centrifuged to a concentration between 1 and $5 \times 10^5$. Approximately 500 µL of binding buffer was added to the suspended cells, after which the medium was supplemented with 5 µL of Annexin V-APC dye solution. After mixing, the cells were incubated in the dark for 10 min at room temperature. Then, the cells were washed twice using FACS buffer and assayed by flow cytometry (CytoFLEX, Beckman Coulter, Krefeld, Germany). Data were analyzed using FlowJo software (BD Biosciences).

## Western blotting assay

Testicular tissues or cells were collected and lysed in cold RIPA lysis buffer (Thermo Scientific, 89900) supplemented with a phosphatase inhibitor (Merck, 524629) and a protease inhibitor phosphatase inhibitor cocktail (Thermo Fisher) with 1 mM PMSF (Beyotime, ST506) for 30 min. The lysates were centrifuged at 12,000 × g for 10 min at 4 °C to remove cell debris. The total protein concentration was measured using a BCA protein assay kit (Thermo Scientific, 23225). To separate the complex mixtures, equal amounts of protein in each group were prepared in SDS sample buffer, subjected to SDS–PAGE and transferred onto nitrocellulose paper (Millipore, Darmstadt, Germany). The PVDF membranes were blocked in 5% nonfat powdered milk and incubated with appropriate primary and secondary antibodies. Proteins were visualized by the enhanced chemiluminescence (ECL) method (Pierce). The signaling intensities were determined from at least three independent blots and were quantified using ImageJ software. The primary antibodies used in this study were as follows: mouse anti-tubulin (Abcam, ab59680, 1:10,000) and rabbit anti-TRPM7 (Affinity, DF7513, 1:100). The secondary antibodies used in this study were as follows: anti-mouse HRP (ZSJB-BIO, zb2305, 1:1000) and anti-rabbit HRP (ZSJB-BIO, zb2301, 1:1000). An ECL kit (Yeasen, 36208ES60) was used on the membrane before film exposure.

## Mitochondrial ROS and membrane potential measurement

After immunostaining for surface markers (lineage makers/CD45/F4/80), single-cell suspensions of testes or cocultures were incubated with MitoSOX™ Red (Invitrogen, M36008) or TMRM (Invitrogen, T668) in PBS at 37 °C for 30 min. The cells were washed twice using PBS and analyzed by flow cytometry. Data were analyzed using FlowJo software and are presented as the mean fluorescence intensity (MFI).

## Lentiviral vector transduction

Lentiviral particles used to label the mitochondria in SLCs or MSCs were purchased from Hanbio Tech (Hanbio, Shanghai, China). The construct was designated pCMV-Mito-DsRed-puro. The SLCs and MSCs (passages 3 or 4) were transfected with the lentivirus and purified using a flow cytometer. TRPM7 shRNA lentiviral particles were purchased from Hanbio Biotechnology. Primary SLCs were plated in a 6-well plate, infected with the TRPM7 lentivirus and purified using puromycin. TRPM7 knockdown was confirmed by western blotting. The following shRNAs were used in this study: mouse *Trpm7* shRNA-1: 5'- CCTGGTATAAGGTCATATTAATT-3', mouse *Trpm7* shRNA-2: 5'-CCT TATCAAACCCTATTGAAT-3', and mouse NC-shRNA: 5'- TTCTCCGA ACGTGTCACGTAA-3'.

## Gene delivery in animal models

AAV viral vectors were prepared by Beijing Tsingke Biotech Co., Ltd. (pAAV-CAG-shTRPM7-mCherry, TST20230904-020-00001). Sequences of shTRPM7: 5'-CCTTATCAAACCCTATTGAAT-3'. Mice were anesthetized by i.p. injection. Using sterile surgical scissors, a single incision was made on the ventral skin and body wall approximately 1.0 cm anterior to the genitals. The testes were pulled out by holding the fat pad. Secure the testis with fine forceps, inject the AAV particles ($8 \times 10^{10}$ gc/testis) into the testicular interstitium using a 33-gauge needle syringe (Hamilton, Switzerland), and then suture the incision. Surgery was performed under aseptic conditions.

## Testicular interstitial fluid exosome extraction

Testicular interstitial fluid was collected as previously reported[62]. Briefly, 2-month-old and 24-month-old C57BL/6 male mice were sacrificed, testes were removed, and the mice were placed in 1 mL of PBS. Then, the mixture was incubated at 4 °C for 45 min and centrifuged at $10000 \times g$ for 15 min at 4 °C. Exosomes derived from testicular interstitial fluid extracted by ultracentrifugation. The above testicular interstitial fluid was centrifuged for 10 min at $2000 \times g$ and 30 min at $10000 \times g$ to remove cell debris. Then, the supernatant was transferred to ultracentrifuge tubes and centrifuged for 2 h at $110,000 \times g$. The pellets were resuspended in 1 mL of PBS, after which the solution was centrifuged for 70 min at $110,000 \times g$ again. The pellets were stored at −80 °C for further analysis.

## Mass spectrometry analysis

The testicular interstitial fluid exosome subcellular fractionations were washed with PBS three times, and protein from each pellet was extracted using UA buffer (8 M urea, 150 mM Tris-HCl pH 8.0) and quantified with a BCA Assay Kit (Bio-Rad, Hercules, CA, USA). LC–MS/MS was carried out at Shanghai Bioprofile Technology Co., Ltd. (China) in the positive-ion mode with an automated data dependent MS/MS analysis. The MS data were analyzed for data interpretation and protein identification against the mouse database from UniProt (UniProt-Reference proteome-*Mus musculus* (Mouse) [10090]−21984-20220804. Fasta), which was sourced from the protein database at https://www.uniprot.org/proteomes/UP000000589. GO terms were grouped into biological process (BP), molecular function (MF), and cellular component (CC) categories. The enriched GO terms were nominally statistically significant at the $p < 0.05$ level. The construction of protein–protein interaction (PPI) networks was conducted by using the online STRING database (https://www.stringdb.org/) with high confidence (0.700) as the threshold.

## RNA-seq analysis

Total RNA was isolated using TRIzol reagent (Fisher Scientific, 15596018), purified using NucleoSpin RNA XS (Takara Bio, U0902A) according to the manufacturers' protocol, and quantified using a Qubit 2.0 (Thermo Fisher Scientific). One nanogram of total RNA per sample was used for the synthesis and amplification of cDNAs. The cDNA libraries were constructed using a TruSeq stranded mRNA kit (Illumina) and sequenced using an Illumina NextSeq 500/550 platform with a NextSeq500/550 Hiqh Output Kit (Illumina). All reads were processed with cutadapt v1.9.1 to remove adaptors and poly-A sequences, and the sequenced fragments were mapped to the mouse genome and assembled using CLC Main Workbench (Qiagen). The DESeq package was used to calculate expression levels for normalized counts. Genes with counts greater than one were included in further analyses. Heatmaps of these genes were generated with the R package. Only genes with an average log-transformed difference greater than 1 and a *p-adj* value less than 0.05 were defined as DEGs. GO analysis was performed with Metascape (www.Metascape.org) using the default parameters.

## Transmission electron microscopy

BMSCs or SLCs were fixed with 2.5% glutaraldehyde (MKBio, 111308) in 0.1 M sodium cacodylate buffer (Electron Microscopy Sciences, 102090-962) and then fixed at 4 °C for preservation. The mitochondria were observed and imaged under TEM at Wuhan Servicebio Technology Co., Ltd. (China). The mitochondria number, width, and length were calculated in BMSCs and SLCs.

## Testosterone test

Orbital blood was collected from mice in each group and centrifuged at $1000 \times g$ for 10 min at 4 °C after being allowed to stand for 1 h to obtain serum. All these serum samples were sent to the Kingmed Diagnostics of Guangzhou to determine the testosterone concentration by ELISA.

## Bone density

The femurs and tibias of the mice were collected and fixed in paraformaldehyde to detect bone mineral density (BMD). Quantitative analysis of the distal femoral metaphysis was performed using microCT. 3D analysis was performed to calculate morphometric parameters at both the lumbar spine (200 slices) and distal femoral metaphysis (100 slices) defining trabecular bone mass.

## Sperm parameter testing

One cauda epididymis was dissected and incubated in 0.5 mL of M2 medium (Sigma, M7167) for 15 min at 37 °C to release sperm. The concentration and motility of sperm were assessed with the Hamilton Thorne's Ceros II system.

## In vitro fertilization (IVF)

Ten-week-old female mice were injected with 10 IU of pregnant mare serum gonadotropin (PMSG, NSHF) and were injected with 10 IU of human chorionic gonadotropin (hCG, NSHF) 48 h later. For IVF, sperm suspensions collected from the cauda epididymis were incubated in a drop of TYH covered with paraffin oil at 37 °C with 5% $CO_2$ for 60 min, and cumulus oocyte complexes collected from superovulated female mice were transferred into a drop of 90 μL of HTF medium (Easy Check, M1130) covered with mineral oil (Sigma–Aldrich, M5310). A 10 μL sperm suspension was transferred to a drop containing oocytes and incubated at 37 °C with 5% $CO_2$. After 6 h, the oocytes with the two pronuclei were scored as fertilized. Mouse embryos were washed in HTF and transferred into KSOM (Easy Check, M1430) medium for further development. We calculated fertilization rates by recording the numbers of 2-cell embryo blastulae.

## Fertility analysis

To detect the fertility of the testicular torsion model in mice, 8-week-old males (saline, $n = 6$; BMSCs group, $n = 6$; SLCs group, $n = 6$) were separately mated with 8-week-old female C57BL/6 mice respectively at a ratio of 1:2. To assess the fertility of old mice, 24-week-old males (control, $n = 6$; SLCs group, $n = 6$; SLCs^shTRPM7 group, $n = 6$) were separately mated with 8-week-old female C57BL/6 mice at a ratio of 1:2. The mating pairs were housed together for four months. The numbers of pups and litters were assessed.

## Statistics and reproducibility

To ensure the reproducibility, all the H&E, immunofluorescence staining, IVF and Western-blot were conducted in at least three biological replicates, unless mentioned otherwise. All data were presented as the mean values ± SDs. All the statistical analyses were performed using GraphPad Prism 9. Unpaired two-tailed Student's t test was used for comparisons between 2 groups, one-way ANOVA was used for comparisons among multiple groups, $p$ values < 0.05 are considered to indicate statistical significance, and a lack of significance is presented in the figures as ns.

## Reporting summary

Further information on research design is available in the Nature Portfolio Reporting Summary linked to this article.

## Data availability

Raw sequencing data for RNA-seq of BMSCs and SLCs was retrieved from Gene Expression Omnibus (GEO) under accession number GSE254881. The mass spectrometry proteomics data have been deposited to the ProteomeXchange Consortium via the PRIDE partner repository with the dataset identifier PXD049210. Source data are provided with this paper.

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

## Acknowledgements

This work was supported by the National Natural Science Foundation of China (T2288101, YJ W), the Natural Science Foundation of Guangdong Province, China (2023A1515010286, MZ), the National Natural Science Foundation of China (81873829, MZ; 81671449, CD), Guangdong Basic and Applied Basic Research Foundation (No. 2021A1515110921, XF), the National Natural Science Youth Foundation of China (82201755, XF), the Guangdong Natural Science Foundation (2023A1515010240, YG). We would like to thank Prof Zhongdao Wu's lab of the Zhongshan School of Medicine, Sun Yat-Sen University for the assistance and suggestions. We are also We are also grateful to the Key Laboratory for Stem Cells and Tissue Engineering, Ministry of Education, Sun Yat-sen University and Shanghai Bioprofile Technology Company Ltd, Laboratory for their technicalassistance.

## Author contributions

Ani Chi and Bicheng Yang designed the experiments. Ani Chi, Hao Dai, Chao Yang, Jie Liu, Menghui Ma,Yanqing Li conducted the experiments. Xinyu Liu performed all bioinformatics analysis. Ani Chi, Chun hua Deng, Xuetao Shi, Min Zhang wrote the manuscript. Hanchao Liu, Jiahui Mo, Yan Liao, Feng Gao, Zhengqing Wang, Yong Gao, Zhihong, Chen and Xin Feng helped with data interpretation and manuscript reviewing. Min Zhang supervised the project.

## Competing interests

The authors declare no competing interests.
