## [Peer Review File · Nature Communications]

REVIEWER COMMENTS

Reviewer #1 (Remarks to the Author):

In the manuscript “Stem Leydig Cells support macrophages immunological homeostasis through intercellular mitochondria transfer in TRPM7 mediated manner in testis”, the authors initially describe the efficiency of transplantation of Stem Leydig Cells (SLCs) over Bone marrow-derived Mesenchymal Stromal Cells (BMSCs) in rescuing the effects of acute testicular ischemia-reperfusion.

Although the idea of the manuscript is fascinating, the data are still preliminary and missing important controls. Below some major comments that need to be addressed by the authors.

1. Result 2.1: SLCs prevent fertility injury. The authors convey the efficiency of SLC transplantation over BMSCs in rescuing testicular injury in the concerned mice model. Although it is evident that SLCs are efficient (Fig. 1), the crucial control of non-injured Wild Type (WT) mice is missing, to prove the point of a rescue effect. This holds true in all the experiments performed across the manuscript.

2. Result 2.1: In a comparison between Fig. 1 O and P, a discrepancy was found. According to the quantification of the number of pups per litter in panel P, the maximum number of pups in the BMSCs group is 2, and that for the SLC group is 4 (based on box-plot with all points shown). However, the representative images in panel O show the discrepancy. The number of pups in BMSC group is 3, and that in SLC group is at least 6. This needs proper justification.

3. Result 2.2: Transplanted SLCs result in recovery of endogenous LCs by promoting differentiation. In this context, the authors do not discuss why the exogenous SLCs are unable to differentiate. Additionally, there are a couple of concerns regarding panels H, I, J, and L of Fig. 2. For H-J, the data will have a profound impact on the readers if the y-axis scale is kept constant for all the three graphs. The authors mention in the text (page 8, lines 9-11) that Fig. 2H represents 10 % of LHR+ cells to also be zsGreen+. This is in fact Fig. 2L. In addition, the y-axis of Fig. 2L needs to be rectified.

4. Result 2.3: This results section has several spelling errors, including the heading. In addition, the title should read “Transplanted SLCs suppress...”. The authors also need to be mindful of double spaces in the text (page 9, lines 24, 28), or lack of spaces between words (page 10, line 6).

5. Result 2.4: Transfer of mitochondria from SLCs. Fig 4C of this section represents the expression of genes related to macrophage recognition, activation, migration and inflammatory response according to the text (page 10, line 28-30), which corresponds to the figure. However, the legend refers to leukocyte related genes. There are several problems with references in this section (page 11). Reference 25 in line 6 does not report microtubules. Reference 29 in line 9 is completely unrelated. Reference 30 in line 11 does not inform about integration of transferred mitochondria into the endogenous organellar network. Additionally, the y-axis of Fig 4F is reported as "Percentage of Mito Tracker Red CMXRos in macrophage. The authors should make it clearer on what is meant by this. Do they mean "Percentage of macrophages with Mito Tracker Red CMXRos? This would make more sense with respect to mitochondrial transfer from SLCs to macrophages.

6. Result 2.4: Besides the qualitative nanotube-like structure observed in Supplementary Fig. 8B, the rationale that the authors provide for looking into nanotubes as a mode of mitochondrial transfer (in lines 4-6 of page 11) is based on their observation of upregulated "microtubule polymerization or depolymerization" signal in SLCs compared to BMSCs. However, their co-culture experiment included a blocker of Actin polymerization (Cytochalasin D), which is not complemented by the initial rationale provided. Additionally, the authors do not provide evidence of microtubule being presented in the nanotube-like connections they observe, which is known to be a cell type-dependent phenotype (not all cells have microtubules in the TNTs they form; also, not all TNTs of the same cell type might have microtubules in them). No experiments were done with Nocodazole, that would interfere with the polymerization of microtubules. In regard to the nanotubes, the representative image provided (Fig 4G) is not convincing. The structure does not represent that of a nanotube. Mitochondrial transfer in this experimental context was analyzed by using MitoTracker CMXRos. In the representative image (Fig. 4G), there seems to be leakiness of the dye, which has previously reported for MitoCMXRos. How do the authors prove that the red signal observed in all acceptor cells in the ROI is not due to leakiness of the dye? Additionally, Fig 4H represents MitoDsRed structures transferred to macrophages in vivo (page 12, lines 12-14). However, the figure mentions MitoCMXRos.

7. Result 2.5: Mitochondrial transfer is upon sensing of ROS by SLCs. In this section, the authors measured mitochondrial ROS. The authors also mention that ROS released by macrophages can affect mitochondrial transfer from SLCs via TRPM7. This section lacks clarity on whether it is mitochondrial superoxide or cytosolic ROS species that is released to activate TRPM7.

8. Result 2.6: SLCs exert anti-aging effects. Page 15, line 9 mentions 24 months old mice. Related figure mentions 22 months old mice. Spelling mistakes need to be taken care of. Additionally, in line 12-13 of page 16, the sentence refers to the number of litters born within a period of 4 months. However, Fig. 6R, S represent the number of pups per litter. This discrepancy needs to be rectified.

In general, the authors need to check spellings of several words in both the main text, and figures.

Reviewer #2 (Remarks to the Author):

This paper investigates the immune modulatory function of stem Leydig cells (SLCs) and their potential for regenerative therapy. Using a testicular torsion model that leads to Leydig cell loss, the authors demonstrate that transplanting SLCs, but not bone marrow MSCs, restores Leydig cell population and hormonal profiles in mice. The study further reveals that transplanted SLCs promote Leydig cell generation from resident progenitors, suppress immune cell infiltration, and reduce macrophage polarization to maintain tissue homeostasis. The mechanism behind SLC's immunomodulatory effects involves nanotube projection from SLCs to macrophages, facilitating the transfer of SLC mitochondria in a TRPM7-dependent manner, thus reducing macrophage inflammatory properties and preventing germ cell loss. Overall, this is an exciting manuscript - raising many fundamental questions for the question.

Comments:

- 1) It would be beneficial to provide information on the extent of Leydig cell loss over time in the testicular torsion model and the kinetics of reversal with and without transplantation. Clarifying this aspect would enhance the understanding of the model's dynamics.
- 2) The authors should discuss why SLCs stimulate endogenous cells to expand and replenish Leydig cells more effectively than the transplanted SLCs. It may be helpful to explore factors such as cell numbers and differentiation probabilities that influence this phenomenon.
- 3) Please specify the timing of SLC or BMSC cell transplantation after testicular torsion event. Does the timing of transplantation affect the percentage of cells that persist at 7 days and their ability to restore tissue homeostasis? Additionally, is the SLC possibly due to immune rejection or cellular crowding, impacting the niche's ability to accommodate new cells.
- 4) Has TRPM7 been previously associated with male infertility? Providing information on the phenotype of SLC conditional knockout (ckO) of TRPM7 mice and whether they exhibit any defects in spermatogenesis or tissue homeostasis after damage would be valuable.

5) Investigate the impact of the age of stem Leydig cells used in transplantation. Comparing young vs. aged stroma using young vs. aged SLCs would be interesting. Additionally, do SLC numbers change with age.

6) Determine the duration of persistence of TRPM7 KD cells in vivo and whether the fertility status at 3 months is a reliable indicator of long-term effects of TRPM7 KD.

7) Perform a thorough proofreading to address the numerous grammatical and typographical errors present in the text. This step will help enhance the clarity and overall quality of the manuscript.

Response to the Reviewers

We thank the reviewers for their thoughtful reading of the manuscript and their insightful and constructive reviews. The reviewers had many positive evaluations. They viewed that our study “the idea of the manuscript is fascinating” (Reviewer #1), and “this is an exciting manuscript” (Reviewer #2). In addition, the Reviewers had many excellent constructive suggestions for improvement. We have performed nearly all of the requested experiments. The revised manuscript includes some new supplemental figures, as well as substantial textual revisions. As a result, we believe the manuscript is much stronger. We wish to take this opportunity to thank the reviewers for their valuable input. Below, we summarize their comments and provide point-by-point responses describing how we have addressed them.

Reviewer #1

In the manuscript “Stem Leydig Cells support macrophage immunological homeostasis through intercellular mitochondria transfer in TRPM7 mediated manner in testis”, the authors initially describe the efficiency of transplantation of Stem Leydig Cells (SLCs) over Bone marrow-derived Mesenchymal Stromal Cells (BMSCs) in rescuing the effects of acute testicular ischemia-reperfusion. Although the idea of the manuscript is fascinating, the data are still preliminary and missing important controls. Below some major comments that need to be addressed by the authors.

Response: We thank the reviewer for these positive comments and kind suggestions. We have performed additional experiments in the revised manuscript as needed. Point-by-point responses to the Reviewer’s comments are provided below.

The major comments:

Point 1: Result 2.1: SLCs prevent fertility injury. The authors convey the efficiency of SLC transplantation over BMSCs in rescuing testicular injury in the concerned mice model. Although it is evident that SLCs are efficient (Fig. 1), the crucial control

of non-injured Wild Type (WT) mice is missing, to prove the point of a rescue effect. This holds true in all the experiments performed across the manuscript.

Response: We appreciate the reviewer's suggestions and agree that noninjured wild-type plants should be included in this study. We have already conducted most of the experiments in the noninjured group (sham group). However, considering space limitations, we mainly emphasized that SLCs have more superior therapeutic potential than BMSCs after testicular torsion; thus, we did not provide sham group data in the original manuscript, as shown in **Rebuttal Fig 1**. As suggested, we added the required results of sham group and reorganized the corresponding data in **Fig.1 to Fig.7 and Supplementary Fig. Fig. 1 to Supplementary Fig. 16**.

Rebuttal Fig 1. Therapeutic effects of SLCs versus BMSCs on testicular function after ischemia–reperfusion. a, Bright field diagram of testicular size (scale bar, 2 mm) and H&E staining of testis samples (scale bar, 50 μ m) obtained from the sham, saline, BMSCs, and SLCs groups on day 28 after testicular torsion. **Red boxes represent the presentation of results in the original manuscript.**

Point 2: Result 2.1: In a comparison between Fig. 1 O and P, a discrepancy was found. According to the quantification of the number of pups per litter in panel P, the maximum number of pups in the BMSCs group is 2, and that for the SLC group is 4 (based on box-plot with all points shown). However, the representative images in panel O show the discrepancy. The number of pups in BMSC group is 3, and that in SLC group is at least 6. This needs proper justification.

Response: We apologize for the mistake as the reviewer pointed out. The representative images were corrected. **The results are presented in Rebuttal Fig. 2 (also see Fig. 1o, p in manuscript).**

Rebuttal Fig 2. SLCs prevent fertility injury in an acute testicular ischemia–reperfusion model. a, Offspring were produced by the sham, saline, BMSCs and SLCs groups. **b,** Quantitative analysis of the number of pups per litter in the sham, saline, BMSCs and SLCs groups. Data are presented as the means \pm SDs, $n=6$ per group, one-way ANOVA was used.

Point 3: Result 2.2: Transplanted SLCs result in recovery of endogenous LCs by promoting differentiation. In this context, the authors do not discuss why the exogenous SLCs are unable to differentiate.

Response: Thank you for your important suggestion. We apologize for not describing the data accurately. The exogenous SLCs were able to differentiate into LCs, as shown in Fig. 2i and 2j. The LHR⁺zsGreen⁺ cells represented the LCs that differentiated from the exogenous SLCs^{zsGreen}. Immunofluorescence staining also revealed that HSD3 β ⁺zsGreen⁺ or CYP11A1⁺zsGreen⁺ (HSD3 β and CYP11A1, the marker of LC) cells were found in the testicular interstitium in the SLCs^{zsGreen} group (Fig 2l and 2 m). These results suggested that a small number of LCs were derived from exogenous SLCs^{zsGreen}. Importantly, we also observed that the transplanted SLCs^{zsGreen} gradually decreased during the early time, and very few cells were found on day 7 after testicular torsion (approximately 0.3%). Most transplanted cells had disappeared (as shown in Fig. 2g, h). which limits the number of transplanted SLCs differentiated into LCs. In line with a previous study, mesenchymal stem cells display

poor survival and engraftment rates in vivo, and their short lifespan could be explained by the stresses encountered after local transplantation, such as nutrient deprivation and proinflammatory cytokines^{1,2}. Notably, a previous study reported that in vivo differentiated stem cells elicit enhanced immunogenicity, thereby increasing their chance for rejection^{3,4}. The state of SLCs differentiation may affect their immunogenicity, leading to only a small number of exogenous LCs surviving in the testis. **Also see the discussion in the revised manuscript as follows:**

“We have already observed the transplanted SLCs gradually decreased during the early time, and very few cells were found on day 7 after testicular torsion (approximately 0.3%), most transplanted cells had disappeared, which limits the number of the transplanted SLCs differentiated into LCs. In line with a previous study, mesenchymal stem cells display poor survival and engraftment rates in vivo, and their short lifespan could be explained by the stresses encountered after local transplantation, such as nutrient deprivation and proinflammatory cytokines^{45,46}. Moreover, a previous study reported that in vivo differentiated stem cells elicit enhanced immunogenicity, thereby increasing their chance for rejection^{47,48}. The state of SLCs differentiation may affect their immunogenicity, leading to only a small number of exogenous LCs surviving in the testis. In our study, we found that transplanted SLCs had the ability to protect ischemic testicular cells from death, which could protect endogenous SLCs from apoptosis. A previous study reported that inflammatory cytokines, including interleukin 6 (IL-6), suppress the differentiation of SLCs, causing a decrease in testosterone production. Other studies have also shown that coculture with aging macrophages, in which the expression of most cytokine genes was upregulated, resulted in a significant reduction in the proliferation of SLCs⁴⁹. We found that although most transplanted cells had disappeared (Fig. 2g, h), the surviving cells attenuated macrophage inflammatory responses, which promoted the subsequent recruitment of leukocytes to the affected testis. An influx of neutrophils and monocytes into the testis occurs, and this phenomenon is gradually resolved following SLC transplantation, which creates a regenerative immune

microenvironment for endogenous SLCs regeneration. In addition to cellular differentiation and close interactions with macrophages, other mechanisms related to paracrine factors may be involved in the therapeutic effects of endogenous SLCs to expand and replenish LCs after testicular torsion.”

Additionally, there are a couple of concerns regarding panels H, I, J, and L of Fig. 2. For H-J, the data will have a profound impact on the readers if the y-axis scale is kept constant for all three graphs.

Response: Thank you for your important suggestion. We apologize for not describing the data accurately. We have revised the figure title as needed.

Rebuttal Fig 3. a, Quantitative analysis of the percentage of BMSCs^{ZsGreen} or SLCs^{ZsGreen} in the saline, BMSCs and SLCs group at groups on days 1, 3 and 7. The data are presented as the means \pm SDs, n = 3 per group, and one-way ANOVA was used.

The authors mention in the text (page 8, lines 9-11) that Fig. 2H represents 10 % of LHR⁺ cells to also be zsGreen⁺. This is in fact Fig. 2L. In addition, the y-axis of Fig. 2L needs to be rectified.

Response: We also thank the reviewer for pointing out this mistake. We have revised the related description of Fig. 2I and rectified the y-axis of Fig. 2I in the revised manuscript as needed. **The results are presented in Rebuttal Fig. 4 (also see Fig. 2j in manuscript).** The rephrased this statement is as follows:

“The ratio of LHR⁺ zsGreen⁻ cells was significantly higher than that of LHR⁺ zsGreen⁺ cells (Fig. 2j)”

“The ratio of zsGreen⁺ or zsGreen⁻ in total LHR⁺ cells ”

Rebuttal Fig 4. a, Quantitative analysis of the percentage of LHR⁺ zsGreen⁺ or LHR⁺ zsGreen⁻ cells in the SLCs group on day 28 after testicular torsion. The data are presented as the means \pm SDs, n = 6 per group, and an unpaired two-tailed Student’s t test was used.

Point 4: Result 2.3: This results section has several spelling errors, including the heading. In addition, the title should read “Transplanted SLCs suppress...”. The authors also need to be mindful of double spaces in the text (page 9, lines 24, 28), or lack of spaces between words (page 10, line 6).

Response: We apologize for this mistake. The rephrased this statement is as follows:

“Transplanted SLCs suppress the inflammatory cascade response in the early stage”

“CD11b⁺Ly6C^{high} monocytes (Fig. 3h, i and Supplementary Fig. 7b) in the SLCs group compared to the BMSCs group”

“SLCs transplantation significantly reduced the CD11b⁺Ly6C^{high} monocyte

(Supplementary Fig. 8a-c) and CD11b⁺Ly6G^{high} (Supplementary Fig. 8a, d, e) neutrophil numbers”

“Macrophages are critical for tissue repair”

Point 5: Result 2.4: Transfer of mitochondria from SLCs. Fig 4C of this section represents the expression of genes related to macrophage recognition, activation, migration and inflammatory response according to the text (page 10, line 28-30), which corresponds to the figure. However, the legend refers to leukocyte related genes.

Response: We apologize for this mistake. We have rephrased this Fig. 4c legend in the revised manuscript as follows:

“Bar graphs showing the expression levels of macrophage homeostasis-related genes.”

“Percentage of macrophages with MitoTracker Red CMXRos”

There are several problems with references in this section (page 11). Reference 25 in line 6 does not report microtubules. Reference 29 in line 9 is completely unrelated. Reference 30 in line 11 does not inform about integration of transferred mitochondria into the endogenous organellar network.

Response: We also thank the reviewer for these important comments for the references in section 2.4. As suggested, we carefully revised the manuscript, especially the references. We have deleted the redundant “Reference 25”, and References 29 and 30 in the revised manuscript have been replaced.

“29. Wang, Y., Li, N., Zhang, X. & Horng, T. Mitochondrial metabolism regulates macrophage biology. J. Biol. Chem. 297, 100904 (2021).”

“30. Jativa, S. et al. Mitochondrial Transplantation Enhances Phagocytic

Function and Decreases Lipid Accumulation in Foam Cell Macrophages. Biomedicines. 10, (2022).”

Additionally, the y-axis of Fig 4F is reported as “Percentage of MitoTracker Red CMXRos in macrophages. The authors should make it clearer on what is meant by this. Do they mean “Percentage of macrophages with Mito Tracker Red CMXRos? This would make more sense with respect to mitochondrial transfer from SLCs to macrophages.

Response: We thank the reviewer for this important input. It does mean what you describe. We have rephrased this Fig. 4f legend in the revised manuscript as follows: **The results are presented in Rebuttal Fig. 5 (also see Fig. 4f in manuscript).** The rephrased this statement is as follows:

Rebuttal Fig 5. a, The percentage of mitochondrial transfer to macrophages in each group was analyzed and graphed. n =3 per group. One-way ANOVA was used.

Point 6: Result 2.4: In addition to the qualitative nanotube-like structure observed in Supplementary Fig. 8B, the rationale that the authors provide for looking into nanotubes as a mode of mitochondrial transfer (in lines 4-6 of page 11) is based on their observation of upregulated “microtubule polymerization or depolymerization” signal in SLCs compared to BMSCs. However, their coculture experiment included a blocker of actin polymerization (cytochalasin D), which is not complemented by the initial rationale provided. Additionally, the authors do not provide evidence of

microtubules being present in the nanotube-like connections they observe, which is known to be a cell type-dependent phenotype (not all cells have microtubules in the TNTs they form; also, not all TNTs of the same cell type might have microtubules in them). No experiments were performed with nocodazole, which interferes with the polymerization of microtubules.

Response: We thank the reviewer for this important input. To address this concern, we designed the following experiment: SLCs were labeled with the red fluorescent probe MitoTracker Red CMXRos and cocultured with macrophages for 24 h, and the coculture system was supplemented with nocodazole. The mitochondrial transfer rate was quantified by flow cytometry. Our results showed that nocodazole significantly decreased the mitochondrial transfer rate, which indicated that microtubules contribute to the formation of TNTs. **The results are presented in Rebuttal Fig. 6 (also see Supplemental Fig. 12f, g of the revised manuscript).**

Rebuttal Fig 6. SLCs transfer mitochondria to macrophages via microtubules. a, Representative flow cytometry profiles showing that SLCs transfer mitochondria to total macrophages with nocodazole treatment for 24 h. **b,** The percentage of mitochondrial transfer to total macrophages in each group was analyzed and graphed. The data are presented as the means \pm SDs, $n = 3$ per group. Unpaired two-tailed Student's t test was used.

In regard to the nanotubes, the representative image provided (Fig 4G) is not convincing. The structure does not represent that of a nanotube. Mitochondrial transfer in this experimental context was analyzed by using MitoTracker CMXRos. In the representative image (Fig. 4G), there seems to be leakiness of the dye, which has

previously been reported for MitoCMXRos. How do the authors prove that the red signal observed in all acceptor cells in the ROI is not due to leakiness of the dye?

Response: Regarding the second concern, we further examined the intercellular interactions in the coculture using field-emission scanning electron microscopy (SEM). SEM image analysis revealed that SLCs and macrophages physically connect via nanotube-like structures. The nanotube arising from SLCs formed contacts with the macrophage membrane (Rebuttal Fig. 6a). **The results are presented in Rebuttal Fig. 7a (see also Supplement Fig. 10c of the revised manuscript).** To rule out the leakiness of the dye, we designed the following experiment: SLCs were transfected with the Mito-dsRed lentivirus and then cocultured with macrophages. Confocal imaging showed that mitochondria from the SLCs were transferred to macrophages through F-actin- and α -tubulin (the structural unit of microtubules)-positive nanotubes and finally internalized into macrophages (Rebuttal Fig. 6b). The results also reconfirmed that microtubules were present in the nanotube-like connections between SLCs and macrophages. **The results are presented in Rebuttal Fig. 7b (see also Fig. 4h of the revised manuscript).**

Rebuttal Fig 7. SLCs and macrophages connect via F-actin and microtubule-positive nanotubes. a, SEM images showing nanotubes (red arrow) between SLCs and macrophages. Scale bar, 10 μ m, the larger scale bar 1 μ m. **b,** Representative confocal microscopy of macrophages cocultured with SLCs labeled with Mito-dsRed. The nanotube was stained with phalloidin and microtubules. Scale

bar, 25 μm ; large scale bar, 5 μm .

Additionally, Fig 4H represents MitoDsRed structures transferred to macrophages in vivo (page 12, lines 12-14). However, the figure mentions MitoCMXRos.

Response: We thank the reviewer for pointing out this mistake in Fig. 4H. We have revised those inappropriate words. **The results are presented in Rebuttal Fig. 8a (see also Fig. 4i of the revised manuscript).**

Rebuttal Fig 8. a, Representative confocal microscopy of SLCs^{Mito-DsRed} mitochondrial transfer to F4/80⁺ (green) macrophages in the interstitial area of the testes on day 1. Scale bar, 50 μm .

Point 7: Result 2.5: Mitochondrial transfer is upon sensing of ROS by SLCs. In this section, the authors measured mitochondrial ROS. The authors also mention that ROS released by macrophages can affect mitochondrial transfer from SLCs via TRPM7. This section lacks clarity on whether it is mitochondrial superoxide or cytosolic ROS species that is released to activate TRPM7.

Response: We thank the reviewer for these comments. To address this concern, we performed additional experiments using the mitochondrial superoxide inhibitor mito-TEMPO to confirm that mitochondrial superoxide is released to activate TRPM7. Flow cytometry revealed that mito-TEMPO treatment significantly reduced the rate of mitochondria that underwent transfer. These results indicated that mitochondrial superoxide influences mitochondrial transfer from SLCs to macrophages. **The results are presented in Rebuttal Fig. 9a, b (see also Supplement Fig. 15h, i of the revised**

manuscript).

Rebuttal Fig 9 Mitochondrial superoxide promotes mitochondrial transfer a, Representative flow cytometry profiles showing that SLCs transfer mitochondria to total macrophages treated with H₂O₂ or mitoTEMOP and H₂O₂ for 24 h. **b,** The percentage of mitochondrial transfer to total macrophages in each group was analyzed and graphed. The data are presented as the means \pm SDs. n = 3 per group. One-way ANOVA was used.

Point 8: Result 2.6: SLCs exert anti-aging effects. Page 15, line 9 mentions 24 months old mice. Related figure mentions 22 months old mice. Spelling mistakes need to be taken care of. Additionally, in lines 12-13 of page 16, the sentence refers to the number of litters born within a period of 4 months. However, Fig. 6R, S represent the number of pups per litter. This discrepancy needs to be rectified.

Response: We apologize for this mistake. We have rephrased these mistakes in the revised manuscript as follows:

“membrane potential was decreased in the testes of 24-month-old mice...”

“The number of pups per litter within 4 months in the SLCs^{shTRPM7} groups was lower than that in the SLCs group.”

Reviewer #2:

This paper investigates the immune modulatory function of stem Leydig cells (SLCs) and their potential for regenerative therapy. Using a testicular torsion model that leads

to Leydig cell loss, the authors demonstrate that transplanting SLCs, but not bone marrow MSCs, restores Leydig cell population and hormonal profiles in mice. The study further reveals that transplanted SLCs promote Leydig cell generation from resident progenitors, suppress immune cell infiltration, and reduce macrophage polarization to maintain tissue homeostasis. The mechanism behind SLC's immunomodulatory effects involves nanotube projection from SLCs to macrophages, facilitating the transfer of SLC mitochondria in a TRPM7-dependent manner, thus reducing macrophage inflammatory properties and preventing germ cell loss. Overall, this is an exciting manuscript - raising many fundamental questions for the question.

Response: We thank the reviewer for the positive comments on both the design and the importance of the work. We would also like to thank the reviewer for the constructive suggestions to investigate fundamental questions for SLCs transplantation. Point-by-point responses to the specific comments from the reviewer are provided below.

The major comments:

Point 1: It would be beneficial to provide information on the extent of Leydig cell loss over time in the testicular torsion model and the kinetics of reversal with and without transplantation. Clarifying this aspect would enhance the understanding of the model's dynamics.

Response: As suggested, we performed additional experiments to investigate the dynamics of LCs reversal over time with BMSCs or SLCs transplantation. The ratio of LCs was analyzed by flow cytometry at 1, 3, 7 and 28 days after testicular torsion. The results showed the proportions of LCs decreased constantly among the whole testis at 1, 3 and 7 days after testicular torsion. SLCs treatment significantly attenuated the declining trend compared to saline and BMSCs group at 1, 3 and 7 days. Importantly, we have already shown that the recovery of LCs was more significant in the SLCs group compared to saline and BMSCs groups on day 28 in Fig 2b and 2c. In summary, the dynamics of the LCs ratio was shown in Rebuttal Fig 4d. These results suggested that SLCs transplantation play a crucial role the maintenance of LCs ratio after testicular

torsion. The results are presented in Rebuttal Fig. 10a-d (see also Supplement Fig. 2a-d).

Rebuttal Fig 10. Transplanted SLCs promote the regeneration of LCs. a, Flow cytometry for detecting the percentage of LHR⁺ cells among the total testicular cells in the saline, BMSCs and SLCs groups on days 1, 3 and 7. b, The dynamic ratio of LCs among the total testicular cells in the saline, BMSCs and SLCs groups on days 1, 3, 7 and 28. Data are presented as the means \pm SDs, n = 6 per group. One-way

ANOVA was used. *p* values indicate statistical significance between SLCs and BMSCs groups.

Point 2: The authors should discuss why SLCs stimulate endogenous cells to expand and replenish Leydig cells more effectively than the transplanted SLCs. It may be helpful to explore factors such as cell numbers and differentiation probabilities that influence this phenomenon.

Response: Thank you for your inspiring query. **This point echoes a similar point raised by reviewer #1's point 3.**

For the first concern, we observed that the transplanted SLCs^{zsGreen} gradually decreased during the early time, and very few cells were found on day 7 after testicular torsion (approximately 0.3%). Most transplanted cells had disappeared (as shown in Fig. 2g, h), which limited the number of transplanted SLCs differentiated into LCs. Moreover, the transplanted SLCs had the ability to protect ischemic testicular cells from death (as shown in Fig. 3e-g). In addition, a previous study reported that inflammatory cytokines, including interleukin 6 (IL-6), suppress the differentiation of SLCs, causing a decrease in testosterone production⁵. Another study also showed that coculture with aging macrophages, in which the expression of most cytokine genes was upregulated, resulted in a significant reduction in the proliferation of SLCs⁶. In our study, although most transplanted cells had disappeared (Fig. 2g, h), the surviving cells could suppress the inflammatory cascade response in the early stage (Fig. 3h-k) and regulate the inflammatory properties of macrophages (Supplementary Fig. 10a-d), which created a regenerative immune microenvironment for endogenous SLC regeneration. Notably, a previous study reported that *in vivo* differentiated stem cells elicit enhanced immunogenicity, thereby increasing their chance for rejection. The state of SLC differentiation may affect their immunogenicity, leading to only a small number of exogenous LCs surviving in the testis. In addition to cellular differentiation and close interactions with macrophages, other mechanisms related to paracrine factors may be involved in the therapeutic effects of endogenous

SLCs to expand and replenish LCs after testicular torsion. **Also see the discussion in the revised manuscript as follows:**

“We have already observed the transplanted SLCs gradually decreased during the early time, and very few cells were found on day 7 after testicular torsion (approximately 0.3%), most transplanted cells had disappeared, which limits the number of the transplanted SLCs differentiated into LCs. In line with a previous study, mesenchymal stem cells display poor survival and engraftment rates in vivo, and their short lifespan could be explained by the stresses encountered after local transplantation, such as nutrient deprivation and proinflammatory cytokines^{45,46}. Moreover, a previous study reported that in vivo differentiated stem cells elicit enhanced immunogenicity, thereby increasing their chance for rejection^{47,48}. The state of SLC differentiation may affect their immunogenicity, leading to only a small number of exogenous LCs surviving in the testis. In our study, we found that transplanted SLCs had the ability to protect ischemic testicular cells from death, which could protect endogenous SLCs from apoptosis. A previous study reported that inflammatory cytokines, including interleukin 6 (IL-6), suppress the differentiation of SLCs, causing a decrease in testosterone production. Other studies have also shown that coculture with aging macrophages, in which the expression of most cytokine genes was upregulated, resulted in a significant reduction in the proliferation of SLCs⁴⁹. We found that although most transplanted cells had disappeared (Fig. 2g, h), the surviving cells attenuated macrophage inflammatory responses, which promoted the subsequent recruitment of leukocytes to the affected testis. An influx of neutrophils and monocytes into the testis occurs, and this phenomenon is gradually resolved following SLC transplantation, which creates a regenerative immune microenvironment for endogenous SLC regeneration. In addition to cellular differentiation and close interactions with macrophages, other mechanisms related to paracrine factors may be involved in the therapeutic effects of endogenous SLCs to expand and replenish LCs after testicular torsion.”

For the second concern, to address whether cell numbers affect the regeneration of LCs, different doses of SLCs^{zsGreen} ranging from 5×10^5 to 2×10^6 cells were transplanted into the testicular interstitium (Rebuttal Fig. 10a). After 28 days, we found that testes were atrophied in the 5×10^5 and 2×10^6 groups compared to the 1×10^6 group (Rebuttal Fig 10b), and the number of LHR⁺ (the total LC population) and LHR⁺zsGreen⁺ (the exogenous population) cells in the whole testis were also the highest in these groups. The 1×10^6 SLC transplantation exerted the best effect on the regeneration of LCs. These results indicated a dose dependence of the therapeutic potential of SLCs, which showed unsatisfactory results with lower and higher doses. **The results are presented in Rebuttal Fig. 11. (see also Supplement Fig. 4a-d).**

Rebuttal Fig. 11 Effect of different doses of SLCs transplantation on LCs recovery. **a**, Experimental timeline. This timeline represents the different doses of SLCs transplantation. **b**, Bright field diagram of testicular size (scale bar, 2 mm) of testis samples obtained from the saline, 5×10^5 , 1×10^6 , and 2×10^6 SLC-treated groups on day 28 after testicular torsion. **c-d**, Quantification of the number of LHR⁺ or LHR⁺zsGreen⁺ cells among the total testicular cells in different groups. The data are presented as the means \pm SDs. $n = 3$ per group. One-way ANOVA was used.

Point 3: Please specify the timing of SLC or BMSC cell transplantation after testicular torsion event.

Response: We appreciate the helpful comments you provided. For the first concern, we apologize for the unclear description of the timing of SLCs or BMSCs transplantation. We injected BMSCs or SLCs immediately after detorsion. We have rephrased the revised manuscript as follows; please also refer to the revised manuscript.

“...injected into the interstitium of the recipient testes after immediately after detorsion...”

Does the timing of transplantation affect the percentage of cells that persist at 7 days and their ability to restore tissue homeostasis?

Response: Regarding the second concern, we thank the reviewer for this suggestion. The timing window is also critical for the efficacy of stem cell therapy to prevent ischemia–reperfusion injury, which influences cell survival and differentiation⁷. However, there is no clearly defined therapeutic time window for cell transplantation after testicular torsion. Since testicular torsion is a urological emergency requiring urgent treatment⁸ and to facilitate practical clinical use by surgeons, we injected BMSCs or SLCs immediately after detorsion in our study.

As suggested, we performed additional experiments to investigate the optimal timing for SLCs transplantation to generate the best therapeutic for LCs recovery. SLCs^{zsGreen} transplantation was performed at 0 h, 12 h, 24 h, and 48 h after testicular torsion (Rebuttal Fig. 11a). On day 7, flow cytometry revealed that the ratio of SLCs^{zsGreen} surviving in the testis after injection was 0.22%, 1.49%, 2.24% and 0.78% in the 0 h, 12 h, 24 h and 48 h transplantation groups, respectively (Rebuttal Fig. 11b, c). The results revealed that the 24 h transplantation group had a significantly higher number of SLCs than all other transplantation groups. On day 28, the ratios of LHR⁺ cells were 0.46%, 1.04%, 1.27%, 1.50%, and 0.84% in the saline, 0 h, 12 h, 24 h, and 48 h groups, respectively (Rebuttal Fig. 11d, e). The above results suggested that the local

microenvironment at 24 h after testicular torsion may be an optimal choice for SLCs transplantation in mice, also expanding our knowledge of SLCs transplantation. To improve the therapeutic effects of SLCs transplantation, strategies to improve resistance to hostile conditions need to be investigated in future studies. **The results are presented in Rebuttal Fig. 12a-e.**

Rebuttal Fig. 12 Effect of SLCs transplantation at different times on LCs recovery. **a**, Experimental timeline. This timeline represents the different timing of SLCs transplantation. **b**, Flow cytometry for detecting the percentage of SLCs^{ZsGreen} among the total testicular cells in the saline, 0 h, 12 h, 24 h and 48 h groups on day 7. **c**, Quantification of the percentage of SLCs^{ZsGreen} among the total testicular cells in the saline, 0 h, 12 h, 24 h and 48 h groups on day 7. The data are presented as the means \pm SDs. $n = 3$ per group, one-way ANOVA was used. **d**, Flow cytometry for detecting the LHR⁺ cells in the saline, 0 h, 12 h, 24 h and 48 h groups on day 28. **e**, Quantification of LHR⁺ cells among the total testicular cells in the saline, 0 h, 12 h, 24 h and 48 h groups on day 28. The data are presented as the means \pm SDs. $n = 3$ per group,

one-way ANOVA was used.

Additionally, is the SLC possibly due to immune rejection or cellular crowding, impacting the niche's ability to accommodate new cells.

Response: The third concern raised by the reviewer is whether the SLCs could impact the niche's ability to accommodate new cells. due to immune rejection or cellular crowding. As tissue-specific mesenchymal stem cells that reside in the testicular interstitium, SLC-based therapies have been reported to be safe in preclinical studies on large animals and can survive for more than one month after transplantation in aging non-human primate testes⁹. It has been reported that mesenchymal stem cells exhibit an immune evasion phenotype due to their low immunogenicity¹⁰. Importantly, transplanted SLCs regulate the inflammatory properties of macrophages and create a regenerative immune microenvironment for LC regeneration. For cellular crowding, fluorescence images showed that transplanted SLCs were distributed in the testicular interstitium, with no apparent cellular crowding (Fig. 2f). We found that the majority of SLCs died within 7 days, and very few cells were found on day 7 after testicular torsion (Fig. 2g and 2 h). Their short lifespan could be explained by the stresses encountered after local transplantation, such as nutrient deprivation and lack of attachment. Importantly, the surviving cells regulate the inflammatory properties of macrophages and create a regenerative immune microenvironment for LC regeneration. On day 28, the recovery of Leydig cells was more significant in the SLCs group than in the saline group and BMSCs group. The dynamics of the LCs ratio also confirmed these results, indicating that SLCs transplantation plays an important role in LCs regeneration, with no impact on the niche's ability to accommodate new cells.

Point 4:

Has TRPM7 been previously associated with male infertility?

Response: Thank you for the constructive comment and interesting perspective. Previous studies have reported that mouse lines in which TRPM7-deficient embryos

cannot develop beyond E7.5^{11,12} and the association between TRPM7 and male infertility have not yet been studied.

Providing information on the phenotype of SLC conditional knockout (ckO) of TRPM7 mice and whether they exhibit any defects in spermatogenesis or tissue homeostasis after damage would be valuable.

Response: We apologize for not being able to measure the phenotype of the TRPM7 knockout mice due to the three-month time limitations to correct manuscripts. To address this concern, we performed experiments using a well-established AAV8-mediated gene expression protocol, AAV8 as a specific efficient vector to mediate gene expression in SLCs¹³. We generated AAV8 vectors that carried shTRPM7 (AAV8-shTRPM7), and 8- to 10-week-old mice were interstitially injected with AAV8-shTRPM7 at doses of 8×10^{10} gc/testis (Rebuttal Fig 12a). Histological analysis of the testes 28 days after vector injection showed the coexpression of mCherry and the SLC-specific marker nestin (Rebuttal Fig 12b). After testicular torsion on day 7, we found that the ratio of CD45⁺F4/80⁺ macrophages was significantly increased in the AAV8-shTRPM7 group compared to the control group (Rebuttal Fig. 12c, d), while the number of germ cells was markedly reduced (Rebuttal Fig. 12e, f). These results suggested that TRPM7 expressed on endogenous SLCs played an important role in the recovery of testis homeostasis after testicular torsion. **The results are presented in Rebuttal Fig. 13a-f (see also Supplement Fig. 17a-f).**

Rebuttal Fig. 13 Knockdown of TRPM7 in endogenous SLCs affects tissue homeostasis after testicular torsion. **a**, Schematic illustration of AAV-shTRPM7 injection into the testicular interstitium of the testes in vivo. **b**, Representative image showing the coexpression of mCherry and the SLC marker nestin. **c**, Flow cytometry for detecting the percentage of CD45⁺F4/80⁺ macrophages among total testicular cells on day 7 after testicular torsion. **d**, Flow cytometry-based percentage of CD45⁺F4/80⁺ macrophages. The data are presented as the means \pm SDs, $n = 3$ per group. Unpaired two-tailed Student's *t* test was used. **e**, Immunostaining of the germ cell marker VASA (red) after testicular torsion in paraffin sections. Scale bars, 50 μ m. **f**, Quantitative analysis of the number of VASA⁺ in seminiferous tubules per section. The data are presented as the means \pm SDs. $n = 3$ per group. Unpaired two-tailed Student's *t* test was used.

Point 5: Investigate the impact of the age of stem Leydig cells used in transplantation. Comparing young vs. aged stroma using young vs. aged SLCs would be interesting.

Response: Thank you for your professional advice. Stem cells play a crucial role in tissue homeostasis and are regulated by their microenvironment¹⁴. The loss of differentiation capacity is considered one of the most important signs of stem cell aging¹⁵. Previous studies have also reported that SLCs decline in their regenerative

potential and ability to differentiate into LCs in the testis, which is associated with a deficiency in testosterone production¹⁶. To further investigate the impact of the age of the SLCs used in transplantation, we performed additional experiments to compare young vs. aged stroma using young vs. aged SLCs as suggested. First, we transplanted young and old SLCs into young EDS-treated mice, an LC-disrupted host animal model, to examine their ability to regenerate damaged tissues in the young niche (Rebuttal Fig 13a). On day 28, flow cytometry revealed that the ratio of young SLCs^{zsGreen} surviving in the EDS mouse testis was significantly higher than that of old SLCs^{zsGreen} (Rebuttal Fig 13b, c). Importantly, the ratio of LHR⁺zsGreen⁺ cells in the young group was significantly higher than that in the old group (Rebuttal Fig 13b, d). These results suggested that aging impairs the capacity of steroidogenic lineage differentiation in SLCs, while the restoration of the young niche is insufficient for rejuvenating the function of exogenous transplanted SLCs, which highlights a key role for age-associated cell-intrinsic defects in SLC aging. **The results are presented in Rebuttal Fig. 14a-d.**

The stem cell niche also undergoes marked changes with aging, which could dictate stem cell behavior^{17,18}. Preclinical and clinical studies have also highlighted that aging significantly impedes the effectiveness of MSC migration and transplantation^{19,20}. Notably, age-related disruption of the testicular microenvironment plays a critical role in LC dysfunction²¹. We then transplanted young or old SLCs into aging mice to examine their ability to regenerate damaged tissues in the aging niche (Rebuttal Fig 13e). Equal amounts of young or old SLCs^{zsGreen} were transplanted into aging mouse testes. On day 28, we found that the ratio of young SLCs^{zsGreen} surviving in the aging mouse testis was significantly higher than that of old SLCs^{zsGreen} (Rebuttal Fig 13f, g), and the ratio of LHR⁺zsGreen⁺ cells in the young group was also significantly higher than that in the old group (Rebuttal Fig 13f, h). Notably, we found that the ratio of young SLCs^{zsGreen} surviving in the young mouse testis (Rebuttal Fig 13i), as well as the ratio of LHR⁺zsGreen⁺ (Rebuttal Fig 13j), was significantly higher than that in the old testis. These results suggested that the aging niche impairs the

capacity for steroidogenic lineage differentiation in SLCs. The results are presented in Rebuttal Fig. 14e-j.

Rebuttal Fig. 14 Characteristics of young or old SLCs transplanted in young or aging mouse testicular interstitium. a, Experimental strategy for the evaluation of SLC survival and differentiation in young mouse testes. **b**, Flow cytometry for

detecting the percentage of SLCs^{ZsGreen} and ZsGreen^{+LHR} cells among the total testicular cells in the young to young, old to young groups on day 28. **c-d**, Quantitative analysis of the SLCs^{ZsGreen} and ZsGreen^{+LHR} cells among the total testicular cells in the young to young old to young groups on day 28. The data are presented as the means \pm SDs, n=3 per group, one-way ANOVA was used. **e**, Experimental strategy for the evaluation of SLCs survival and differentiation in aging mouse testes. **f**, Flow cytometry for detecting the percentage of SLCs^{ZsGreen} and ZsGreen^{+LHR} cells among the total testicular cells in the young to old and old to old groups on day 28. **c-d**, Quantitative analysis of the SLCs^{ZsGreen} and ZsGreen^{+LHR} cells among the total testicular cells in the young to old and old to old groups on day 28. The data are presented as the means \pm SDs, n = 3 per group, one-way ANOVA was used. **i**, Quantitative analysis of the SLCs^{ZsGreen} among the total testicular cells in the young to young and young to old groups on day 28. The data are presented as the means \pm SDs, n = 3 per group. Unpaired two-tailed Student's t test was used. **j**, Quantitative analysis of the ZsGreen^{+LHR} cells among the total testicular cells in the young to young and young to old groups on day 28. The data are presented as the means \pm SDs, n=3 per group. Unpaired two-tailed Student's t test was used.

Additionally, do SLCs numbers change with age?

Response: We thank the reviewer for these comments. Our previous studies have reported that the number of SLCs gradually decreased with age in the total population of the interstitial compartment, with a significant peak on postnatal day 7, which then gradually decreased by day 90²². As suggested, we also performed additional flow cytometry analysis of the SLCs ratio in aging mice. The results showed that the ratio of SLCs was significantly decreased in aging mice compared to young mice. **The results are presented in Rebuttal Fig 15. a, b.**

Rebuttal Fig. 15 Characteristics of the percentage of SLCs in young or aging mice. **a**, Flow cytometry for detecting the percentage of SLCs cells in mouse testes from different age groups. **b**, Quantification of the percentage of SLCs in mouse testes from different age groups. The data are presented as the means \pm SDs, $n=3$ per group. Unpaired two-tailed Student's *t* test was used.

Point 6:

Determine the duration of persistence of TRPM7 KD cells in vivo and whether the fertility status at 3 months is a reliable indicator of long-term effects of TRPM7 KD.

Response: We thank the reviewer for these important suggestions. As suggested, we performed additional experiments to examine the effects of TRPM7 knockdown on fertility status in aging mice at 3 months. Equal amounts of SLCs^{zsGreen} and SLCs^{zsGreen/shTRPM7} were transplanted into the testicular interstitium. After 3 months, flow cytometry analysis revealed that the ratio of SLCs^{zsGreen} was significantly higher than that of SLCs^{zsGreen/shTRPM7}. In addition, sperm counts and sperm motility in the epididymis were notably higher in the SLCs^{zsGreen} group than in the SLCs^{zsGreen/shTRPM7} group. In vitro fertilization (IVF) further confirmed the improved sperm quality, and we found significantly higher rates of 2-cell embryos and blastocysts in the SLCs^{zsGreen} group. These findings suggested that knockdown of TRPM7 impairs the therapeutic effect of SLCs to promote fertility recovery at 3 months in aging mice. **The results are presented in Rebuttal Fig. 16a-f (see also Supplement Fig. 20a-f).**

Rebuttal Fig. 16 The therapeutic effects of SLCs versus SLCs^{shTRPM7} on the fertility status of aging mice at 3 months. **a**, Schematic illustration of SLCs^{zsGreen} or SLCs^{zsGreen/shTRPM7} injection into the testicular interstitium in aging mice. **b**, Flow cytometry for detecting the percentage of zsGreen⁺ cells in total testicular cells after three months. **c**, Quantification analysis of the percentage of zsGreen⁺ cells in total testicular cells after three months. The data are presented as the means \pm SDs. $n = 3$ per group. Unpaired two-tailed Student's t test was used. **d-f**, Sperm counts and sperm motility in the SLCs and SLCs^{shTRPM7} groups. PR: progressive motile, NR: nonprogressive motile, IM: immotilitile. The data are presented as the means \pm SDs, $n=3$ per group, one-way ANOVA was used. **g**, The trilinear table shows all embryo injection, two-cell embryo and blastocyst data in the saline, SLCs and SLCs^{shTRPM7} groups. **h-i**, Bright field diagram of the 2-cell, blastocyst stages among the saline,

SLCs and SLCs^{shTRPM7} groups. The arrows indicate normal developing embryos. Scale bar, 200 μm .

Point 7:

Perform a thorough proofreading to address the numerous grammatical and typographical errors present in the text. This step will help enhance the clarity and overall quality of the manuscript.

Response:

We apologize for any errors that were due to our oversight during the preparation of the original manuscript. Accordingly, the revised manuscript was proofread by native English professionals from American Journal Experts. We have emphasized correcting articles and singular/plural forms in the revised manuscript, and we believe that the revised manuscript would be much more readable.

References

1. Yu, S., Yu, S., Liu, H., Liao, N. & Liu, X. Enhancing mesenchymal stem cell survival and homing capability to improve cell engraftment efficacy for liver diseases. *Stem Cell Res. Ther.* **14**, 235 (2023).
2. Barrere-Lemaire, S. et al. Mesenchymal stromal cells for improvement of cardiac function following acute myocardial infarction: a matter of timing. *Physiol. Rev.*, (2023).
3. Abu-El-Rub, E. et al. Hypoxia-induced increase in Sug1 leads to poor post-transplantation survival of allogeneic mesenchymal stem cells. *Faseb J.* **34**, 12860-12876 (2020).
4. Swijnenburg, R. J. et al. Embryonic stem cell immunogenicity increases upon differentiation after transplantation into ischemic myocardium. *Circulation.* **112**, I166-I172 (2005).
5. Wang, Y. et al. Interleukin 6 inhibits the differentiation of rat stem Leydig cells. *Mol. Cell. Endocrinol.* **472**, 26-39 (2018).
6. Shao, J. et al. Effects of aging and macrophages on mice stem Leydig cell proliferation and differentiation in vitro. *Front. Endocrinol.* **14**, 1139281 (2023).
7. Rosenblum, S. et al. Timing of intra-arterial neural stem cell transplantation after hypoxia-ischemia influences cell engraftment, survival, and differentiation. *Stroke.* **43**, 1624-1631 (2012).
8. Jativa, S. et al. Mitochondrial Transplantation Enhances Phagocytic Function and Decreases Lipid Accumulation in Foam Cell Macrophages. *Biomedicines.* **10**, (2022).
9. Xia, K. et al. Restorative functions of Autologous Stem Leydig Cell transplantation in a Testosterone-deficient non-human primate model. *Theranostics.* **10**, 8705-8720 (2020).
10. Xia, C. & Cao, J. Imaging the survival and utility of pre-differentiated allogeneic MSC in ischemic heart. *Biochem. Biophys. Res. Commun.* **438**, 382-387 (2013).
11. Schutz, A. et al. Trophectoderm cell failure leads to peri-implantation lethality in Trpm7-deficient mouse embryos. *Cell Reports.* **37**, 109851 (2021).
12. Jin, J. et al. Deletion of Trpm7 disrupts embryonic development and thymopoiesis without altering Mg²⁺ homeostasis. *Science.* **322**, 756-760 (2008).
13. Xia, K. et al. AAV-mediated gene therapy produces fertile offspring in the Lhcgr-deficient mouse model of Leydig cell failure. *Cell Rep. Med.* **3**, 100792 (2022).

14. Brunet, A., Goodell, M. A. & Rando, T. A. Ageing and rejuvenation of tissue stem cells and their niches. *Nat. Rev. Mol. Cell Biol.* **24**, 45-62 (2023).
15. Roobrouck, V. D., Ulloa-Montoya, F. & Verfaillie, C. M. Self-renewal and differentiation capacity of young and aged stem cells. *Exp. Cell Res.* **314**, 1937-1944 (2008).
16. Yao, S. et al. Nestin-dependent mitochondria-ER contacts define stem Leydig cell differentiation to attenuate male reproductive ageing. *Nat. Commun.* **13**, 4020 (2022).
17. Chakkalakal, J. V., Jones, K. M., Basson, M. A. & Brack, A. S. The aged niche disrupts muscle stem cell quiescence. *Nature.* **490**, 355-360 (2012).
18. Matteini, F., Mulaw, M. A. & Florian, M. C. Aging of the Hematopoietic Stem Cell Niche: New Tools to Answer an Old Question. *Front. Immunol.* **12**, 738204 (2021).
19. Fabian, C. et al. Distribution pattern following systemic mesenchymal stem cell injection depends on the age of the recipient and neuronal health. *Stem Cell Res. Ther.* **8**, 85 (2017).
20. Zhang, K. et al. The impact of recipient age on the effects of umbilical cord mesenchymal stem cells on HBV-related acute-on-chronic liver failure and liver cirrhosis. *Stem Cell Res. Ther.* **12**, 466 (2021).
21. Curley, M. et al. A young testicular microenvironment protects Leydig cells against age-related dysfunction in a mouse model of premature aging. *Faseb J.* **33**, 978-995 (2019).
22. Jiang, M. H. et al. Characterization of Nestin-positive stem Leydig cells as a potential source for the treatment of testicular Leydig cell dysfunction. *Cell Res.* **24**, 1466-1485 (2014).

REVIEWERS' COMMENTS

Reviewer #1 (Remarks to the Author):

The authors replied to all my comments and in my opinion the paper is now acceptable for publication

Reviewer #2 (Remarks to the Author):

The authors have address the key points raised by reviewers.